# Evaluation of Biases in mid-to-high latitudes Surface Snowfall and Cloud Phase in ERA5 and CMIP6 using Satellite Observations

Franziska Hellmuth[1], Tim Carlsen[1], Anne Sophie Daloz[2], Robert Oscar David[1], Haochi Che[1], and Trude Storelvmo[1]

[1]Department of Geosciences, University of Oslo, Blindernveien 31, 0371, Oslo, Norway
[2]Center for International Climate Research (CICERO), Gaustadalleen 21, 0349, Oslo, Norway

**Correspondence:** Franziska Hellmuth (franziska.hellmuth@geo.uio.no)

**Abstract.** Supercooled Liquid-Containing Clouds (sLCCs) play a significant role in Earth's radiative budget and the hydrological cycle, especially through surface snowfall production. Evaluating state-of-the-art climate models with respect to their ability to simulate the frequency of occurrence of sLCCs and the frequency with which they produce snow is, therefore, critically important. Here, we compare these quantities as derived from satellite observations, reanalysis datasets, and Earth System
Models from Phase 6 of the Coupled Model Intercomparison Project (CMIP6) and find significant discrepancies between the data sets for mid and high latitudes in both hemispheres. Specifically, we find that the ERA5 reanalysis and ten CMIP6 models consistently overestimate the frequency of sLCCs and snowfall frequencies from sLCCs compared to CloudSat-CALIPSO satellite observations. The biases are very similar for ERA5 and the CMIP6 models, which indicates that the discrepancies in cloud phase and snowfall stem from differences in the representation of cloud microphysics rather than the representa-
tion of meteorological conditions. This, in turn, highlights the need for refinements in the models' parameterizations of cloud microphysics in order for them to represent cloud phase and snowfall accurately. The thermodynamic phase of clouds and precipitation has a strong influence on simulated climate feedbacks and, thus, projections of future climate. Understanding the origin(s) of the biases identified here is, therefore, crucial for improving the overall reliability of climate models.

## 1 Introduction

Snowfall and snow cover have a significant impact on the Earth's energy budget and the hydrological cycle, especially in mid- and high-latitudes, and thereby strongly impact ecosystems and human societies. Snow cover influences vegetation growth, animal populations, and ecosystem processes, while also impacting economic activities, infrastructure, and health (Callaghan et al., 2011; Bokhorst et al., 2016). Indeed, snowfall is also an important water resource globally (Barnett et al., 2005) and snow cover increases the surface albedo, reducing the absorption of incoming solar energy (Zhang, 2005). However, heavy snowfall
events have the potential to negatively impact local communities and infrastructure (Eisenberg and Warner, 2005; Scott et al., 2008; Fox et al., 2023).

Cloud phase determines the formation process of snow. Snow can originate from pure ice clouds as well as supercooled liquid-containing clouds (sLCCs), where both supercooled water droplets and ice crystals coexist at temperatures between

$-40°C$ and $0°C$ (Jiusto and Weickmann, 1973; Korolev et al., 2017). These sLCCs dominate at latitudes higher than $45°$ and cover up to $20\% - 30\%$ of the Earth depending on the season (Warren et al., 1988; Matus and L'Ecuyer, 2017). The cloud phase affects not only the rate and intensity of snowfall but also the microphysical properties of the snowflakes that reach the ground (Jiusto and Weickmann, 1973; Liu, 2008). Clouds reflect solar radiation (cooling the surface) and trap terrestrial radiation (warming the surface). The balance between the two effects is partly influenced by the phase composition of the clouds due to distinct scattering properties of liquid and ice (Shupe and Intrieri, 2004; Ehrlich et al., 2009; Cesana et al., 2012). For otherwise similar properties, a cloud with more liquid reflects more solar radiation back to space. Therefore, representing the cloud phase correctly in Earth System Models (ESMs) has substantial implications for the simulation of cloud radiative properties (Matus and L'Ecuyer, 2017), especially in future climate projections with a warmer climate. This represents an important component of the cloud feedback, i.e., how clouds respond to changes in surface air temperature and how these changes, in turn, influence temperature. The cloud feedback is, at present, the most uncertain physical climate feedback, significantly contributing to the spread of climate sensitivity in ESMs (Zelinka et al., 2020). Among the most poorly understood cloud feedbacks is the one associated with cloud phase changes (Bjordal et al., 2020; Zelinka et al., 2020).

Atmospheric warming can lead to cloud phase changes through two different pathways: less cloud ice due to higher temperatures or more ice due to potentially more ice-nucleating particles (INPs). INPs are microscopic particles present in the air that serve as the starting point of ice crystal formation in clouds (DeMott et al., 2010). Several studies, including Murray et al. (2021) showed that increased temperatures, especially in polar regions, have caused a shift of mixed-phase clouds towards higher latitudes and altitudes due to the ice reduction in the atmosphere in these regions. The shift in cloud phase towards more liquid and less ice leads to a reduction in the fraction of precipitation falling as snow in previously snowy areas in the Northern Hemisphere (NH), but snowfall events will happen further north in the future (Chen et al., 2020). This shift would result in an expected decrease in snowfall events and duration of the snowfall season for most regions in the NH (Danco et al., 2016; Chen et al., 2020). However, potentially partly counteracting this, the reduction in Arctic sea ice may also facilitate local emission of INPs (Carlsen and David, 2022). Furthermore, the well-established general increase in total precipitation with warming may also lead to increased snowfall in some regions (Douville et al., 2023). Hence, cloud phase, snowfall amounts, and snowfall frequency of occurrence will likely change with warming in ways that are both complex and currently poorly understood (Danco et al., 2016; Chen et al., 2020; Quante et al., 2021).

There are several processes governing snow formation within sLCCs. The transition from a fully liquid to a completely frozen cloud can follow various pathways (Costa et al., 2017), driven by differences in saturation vapor pressures of ice and liquid at temperatures below $0°C$ (Wegener, 1912; Bergeron, 1928; Findeisen, 1938), where ice crystals will grow at the expense of supercooled liquid water (Korolev and Mazin, 2003; Korolev, 2007; Storelvmo and Tan, 2015; Korolev et al., 2017). This phenomenon is called the Wegener-Bergeron-Findeisen (WBF) process and can lead to the rapid growth of ice crystals, which eventually fall out as snow. Increasing INP concentrations can cause cloud glaciation, amplified by the WBF process, increasing the precipitation at the surface and consequently shortening the cloud lifetime (Lohmann and Diehl, 2006; Rosenfeld et al., 2011; Storelvmo and Tan, 2015). Furthermore, ice crystals can be introduced into this sLCC by either falling into it from above (e.g. Proske et al., 2021) or by turbulent mixing.

The microphysical processes described above occur on scales smaller than the grid resolution used in ESMs and their influence on cloud macroscopic properties must therefore be parameterized. Cloud microphysical parameterizations in ESMs are known to often be overly crude and simplistic. For example, ESMs generally assume that liquid and ice are uniformly mixed within a grid box, while observations have shown that sLCCs typically consist of cloud pockets exclusively composed of liquid or ice (Korolev and Milbrandt, 2022). Additionally, the treatment of primary ice production has an influence on the amount of supercooled liquid water in climate models (Vergara-Temprado et al., 2018). Only a small number of ESMs include parameterizations of primary ice production that depend on the presence of aerosols with the ability to act as INPs, while the majority include simpler parameterizations that rely only on temperature. Cesana et al. (2015) found that models with more complex microphysics (e.g., prognostic ice and liquid water content, heterogeneous freezing, riming, accretion, and the WBF process) tend to provide a more accurate representation of the ice phase. Similarly, Komurcu et al. (2014) noted that the variability in cloud phase among models was influenced by the specific ice nucleation scheme used and the representation of other microphysical processes associated with ice.

ESMs have previously been shown to not accurately represent cloud phase by often underestimating liquid and overestimating ice, compared to satellite measurements. This has particularly been the case for high-latitude regions (Komurcu et al., 2014; Cesana et al., 2015; Tan and Storelvmo, 2016; Kay et al., 2016; McIlhattan et al., 2017; Bruno et al., 2021; Shaw et al., 2022). However, most studies to date were conducted based on the previous generation of ESMs (Coupled Model Intercomparison Project 5 (CMIP5), Taylor et al., 2012), and a wide range of different cloud phase metrics have been applied in the past, often without a clear strategy that allows like-for-like comparison with satellite observations. In other words, uncertainties remain in understanding the complex processes governing cloud phase and their representation in ESMs (Komurcu et al., 2014). Because cloud phase and snowfall are tightly linked through the processes outlined earlier, any inaccuracies in representing cloud phase could lead to biases in the simulation of snow growth, formation, and the precipitation reaching the ground in solid or liquid form (Mülmenstädt et al., 2021; Stanford et al., 2023). It is important to note, however, that while such biases in cloud phase representation might exist, other compensating model biases could nevertheless lead to accurate precipitation simulations.

CloudSat (Stephens et al., 2002) and Cloud-Aerosol Lidar and Infrared Pathfinder Satellite Observation (CALIPSO, Winker et al., 2010), flying in the afternoon-train constellation, provide global estimates of cloud properties and snowfall since 2006. CloudSat is equipped with a Cloud Profiling Radar (CPR) that detects large cloud and precipitation particles, while CALIPSO with its Cloud-Aerosol Lidar with Orthogonal Polarization (CALIOP, Winker et al., 2007), can among other things, determine the phase of cloud layers. The CloudSat-CALIPSO constellation overpasses the same regions approximately every 16 days, providing long-term periodic monitoring of cloud and snowfall characteristics across the globe. Previous studies have contributed to a better understanding of the uncertainties associated with satellite measurements of clouds and precipitation (e.g., Stephens and Kummerow, 2007; Hiley et al., 2011). For example, Stephens and Kummerow (2007) identified two primary sources of uncertainty in retrieval methods, errors in distinguishing between cloudy and clear sky scenes and between precipitating and non-precipitating clouds. Furthermore, the forward models used are highly sensitive to their input parameters, particularly the radiative transfer and atmospheric models. Hiley et al. (2011) demonstrated that snowfall retrievals are also influenced by retrieval assumptions and the use of different ice particle models, which can significantly affect the estimated

snowfall rates. Despite limitations, the value of satellite data like CloudSat and CALIPSO for validating reanalysis and ESM

data, which both rely on parameterized microphysics (Forbes and Ahlgrimm, 2014; McIlhattan et al., 2017; Milani et al., 2018; Edel et al., 2020; Daloz et al., 2020) is well established. Previous studies suggest that there is a notable difference in snowfall estimates between ESMs and satellite observations. For example, Heymsfield et al. (2020) found that the Met Office Unified Model and the Community Atmosphere Model 6 produce double the amount of snowfall relative to satellite observations. Additionally, Roussel et al. (2020) discovered that the ensemble median of CMIP5 experiments tend to show a positive bias

in snowfall rates compared to the CloudSat average. Of particular relevance for the analysis presented here is the study by McIlhattan et al. (2017), who investigated the causes and impacts of liquid-containing cloud (LCC) biases in the Arctic region. They found that the Community Earth System Model Large Ensemble (CESM-LE) underestimates the frequency of Arctic LCCs compared to observations. Moreover, the CESM-LE overestimated the snowfall frequency from these clouds, possibly indicating an overactive WBF process leading to too frequent snowfall and too short cloud lifetime (McIlhattan et al., 2017).

In a comparable study of the Southern Hemisphere (SH), Roussel et al. (2020) showed a weak annual cycle of snowfall rates in the Antarctic plateau regions and found that the CMIP5 and CMIP6 (Eyring et al., 2016) simulations tended to overestimate the average precipitation rate in Antarctica. Despite the improvements in surface temperature representation from CMIP5 to CMIP6, there is no corresponding improvement in the representation of large-scale mean precipitation rate and seasonality in the region (Roussel et al., 2020). Precipitation errors in ESMs are not primarily driven by first-order physical links but by

atmospheric circulation and cloud microphysics (Roussel et al., 2020). Antarctica and the surrounding ocean are also known to be a region in which ESMs have large biases, which are thought to be attributable to cloud phase biases (e.g. Vergara-Temprado et al., 2018). These persistent discrepancies between ESMs and observations further highlight the importance of understanding the link between cloud phase and solid precipitation on different scales and improving the representation of cloud phase in ESMs.

Reanalysis products employ numerical weather prediction models that assimilate various observations to generate a continuous spatial and temporal dataset. The European Center for Medium-Range Weather Forecasts (ECMWF) reanalysis datasets use data collected from satellites, weather stations, and ocean buoys to combine with a numerical weather prediction model to provide a comprehensive view of the global atmospheric climate. Nonetheless, the precision in depicting cloud and surface snowfall relies on the underlying model and the assimilated dataset (Boisvert et al., 2018; Daloz et al., 2020; Boisvert et al.,

2020). Challenges may arise due to limited spatial resolution and sparse observations, particularly in remote and complex topographical regions (Boisvert et al., 2018; Daloz et al., 2020; Boisvert et al., 2020). Wang et al. (2019) studied snowfall within two ECMWF reanalysis data sets, ERA-Interim (Simmons et al., 2007; Dee et al., 2011) and ERA5 (Hersbach et al., 2020). The latter includes higher horizontal resolution, improvements to the numerical model, improved data assimilation, and different cloud physics schemes (Forbes and Ahlgrimm, 2014). Wang et al. (2019) showed that ERA5 led to a better representation

of snowfall than ERA-Interim. Nevertheless, Milani et al. (2018) concluded that ERA-Interim reanalysis data produce mean annual snowfall patterns similar in magnitude to CloudSat over Antarctica. In the Arctic, the ERA-Interim data qualitatively represented the interannual snowfall rates and seasonal cycle well but underestimated high snowfall rates significantly during summer and overestimated weak snowfall rates over open water compared to CloudSat (Edel et al., 2020). Seasonal biases

tended to be higher in colder months when heavier snowfall occurred (Edel et al., 2020). Boisvert et al. (2020) compared different reanalysis products including ERA-Interim and ERA5, and focused on snowfall in the Southern Ocean. They found similar spatial and interannual snowfall patterns, where ERA-Interim produced the least snowfall compared to ERA5 and the other reanalysis datasets. Sea-ice representation, atmospheric moisture content, temperature, cloud microphysics schemes, and data assimilation used in the reanalysis were identified to all contribute to the disagreement on snowfall magnitude at lower latitudes (north of the Antarctic continent, Boisvert et al., 2020). They also showed that the difference between subsequent reanalysis iterations is due to these factors, and model resolution changes. However, it is challenging to attribute the differences further to specific factors.

This study adds to the previous literature discussed above by connecting cloud phase and surface snowfall in ERA5 reanalysis, CMIP6 models, and CloudSat-CALIPSO observations through analyses of the frequency of occurrence of sLCCs and associated surface snowfall in these data sets. The frequency of sLCC occurrence ($f_{sLCC}$) represents the fraction of time (in percent) that sLCCs are observed, while the frequency of snowfall from sLCC ($f_{snow}$) denotes the percentage of time that sLCCs are snowing. This information is crucial for understanding when and where sLCCs produce snowfall and is vital for predicting and modeling precipitation patterns in high-altitude and latitudinal areas. By studying the $f_{snow}$, valuable insights into the occurrence of snowfall can be gained, further enhancing our understanding of its role in the water cycle, as well as the representation of the WBF process in reanalysis and ESMs.

Another novel contribution is the assessment of cloud phase biases based on a relatively new cloud phase metric that aims for a like-for-like comparison with satellite retrievals to the extent possible, and the relation of these biases to associated snowfall biases in reanalysis and ESMs. The study area spans the latitudinal ranges of 45°N - 82°N and 45°S - 82°S. We select these regions because they cover the mid-latitudes and polar areas where mixed-phase clouds and surface snowfall are predominantly observed (Warren et al., 1988; Komurcu et al., 2014; Korolev et al., 2017; Matus and L'Ecuyer, 2017; Chen et al., 2020). The central research question of this study is: What are the cloud phase biases in the NH and SH mid-to-high latitudes in ERA5 and *historical* simulations from CMIP6 with respect to the relatively new cloud phase metrics, and how do these biases relate to snowfall biases? By addressing this research question, we aim to improve our understanding of cloud phase representation and its connection to snowfall and ultimately contribute to the advancement of climate modeling and prediction.

In the subsequent sections, we will investigate the frequency and geographic distribution of sLCCs and associated snowfall in order to distinguish their contribution to the overall snowfall patterns. Section 2 of this paper includes a description of the datasets and methodologies utilized in this study. Section 3 presents the results for the $f_{sLCC}$ and $f_{snow}$ for the NH and SH mid-to-high latitudes, between 2007 and 2010. Section 4 is a discussion on CloudSat-CALIPSO and the use of different time sampling for the various datasets. In Section 5, the connection between cloud phase and snowfall biases is discussed and gives possible next steps for future studies.

**Table 1.** Overview of variable and original data field name, units, and data properties for CloudSat-CALIPSO, ERA5, and CMIP6 datasets. CMIP6 model variables represent variable short names.

| Variable | Data field name | Units | Variable property |
|---|---|---|---|
| **CloudSat-CALIPSO** | | | |
| Cloud phase flag | CloudPhase | ice, mixed, liquid | from: 2B-CLDCLASS-LIDAR |
| 2-metre temperature | Temperature_2m | K | from: ECMWF-AUX |
| Surface snowfall rate | snowfall_rate_sfc | $mmh^{-1}$ | from: 2C-SNOW-PROFILE |
| **ERA5** | | | |
| Total column cloud liquid water | tclw | $kgm^{-2}$ | Instantaneous |
| Total column rainwater | tcrw | $kgm^{-2}$ | Instantaneous |
| 2 metre temperature | 2t | K | Instantaneous |
| Mean snowfall rate | msr | $kgm^{-2}s^{-1}$ | Mean rate |
| **CMIP6** | | | |
| Mass Fraction of Cloud Liquid Water | clw | $kgkg^{-1}$ | model level |
| Near-Surface Air Temperature | tas | K | single level |
| Snowfall Flux | prsn | $kgm^{-2}s^{-1}$ | single level |
| Pressure on Model Half-Levels | phalf | Pa | model half level |
| Grid-Cell Area for Atmospheric Grid Variables | areacella | $m^2$ | single level |

## 2  Data and Methods

Our primary emphasis is on sLCCs in the mid-to-high latitudes and understanding how frequently and where they produce snowfall. We utilize CloudSat-CALIPSO satellite observations, ERA5 reanalysis, and CMIP6 simulations to assess biases both in the spatial distribution and frequency of occurrence of sLCCs as well as their respective snowfall frequency. This section introduces the different data sets and presents the statistical methods used for the cloud phase and snowfall analysis. Table 1 lists the units and data properties for the variables used from CloudSat-CALIPSO, ERA5, and CMIP6, respectively.

### 2.1  CloudSat-CALIPSO Satellite Retrievals

In our investigation of the relationship between sLCCs and surface snowfall biases, we rely on satellite observations from CloudSat and CALIPSO for the comparison with ERA5 and CMIP6 models. Specifically, we use Release 5 (R05) versions of the 2B-CLDCLASS-LIDAR product (Sassen et al., 2008) for cloud phase determination, the 2C-SNOW-PROFILE product (Wood and L'Ecuyer, 2018) for the surface snowfall rate estimation, and 2m temperatures from the ECMWF-AUX data set that contains ancillary ECMWF state variable data interpolated to each vertical radar bin (see Table 1).

The 2B-CLDCLASS-LIDAR product utilizes the CALIOP lidar and the CPR on CloudSat and their different sensitivities to liquid droplets and ice crystals in order to retrieve the phase of a cloudy layer. The algorithm uses a temperature-dependent radar reflectivity threshold (Zhang et al., 2010), the integrated attenuated lidar backscattering coefficient, as well as cloud base and top temperatures from atmospheric reanalysis data to discriminate between ice, mixed, or liquid water clouds (Wang and Sassen, 2019). In accordance with McIlhattan et al. (2017), we focus on the phase of the bottom cloud layer within the atmospheric column and neglect the cloud phase quality flag, irrespective of the confidence level indicated. The rationale for the latter is the lidar's ability to robustly detect any liquid (liquid or mixed), and a low confidence value for the phase determination could stem from the uncertainty in the distinction between purely liquid clouds and mixed-phase clouds. This is particularly true for the frequent structure of polar mixed-phase clouds with the liquid layer at cloud top (McIlhattan et al., 2017).

To estimate the surface snowfall rate, we use the 2C-SNOW-PROFILE product (Wood and L'Ecuyer, 2018), which is based on an optimal estimation algorithm to retrieve profiles of parameters of the snow size distribution. The optimal estimation uses the radar reflectivity profile of the snow layer, ancillary meteorological information, and assumptions about snow microphysical and scattering properties. As the CPR cannot reliably measure near-surface reflectivities due to ground clutter, the surface snowfall rate is estimated based on the lowest clutter-free radar bin (Wood et al., 2014). The truncation height of the snow profile due to ground clutter is surface dependent (2 range bins above surface over ocean ($\sim 500$m), 4 range bins over land and sea ice ($\sim 1000$m), Wood et al., 2014).

We focus on the years 2007 to 2010 before CloudSat switched to daytime-only operations due to a battery malfunction. Furthermore, we exclude September 2008 and December 2009 due to insufficient CALIOP data and a CloudSat battery failure (Keys, 2010). We aggregate the profile-by-profile data to a horizontal grid with a resolution of $3.75°$x$1.9°$ for each month. Subsequently, we calculate the monthly $f_{sLCC}$ and $f_{snow}$ as described in Sect. 2.4.

The sea ice concentration (SIC) data used in this study is obtained from the Institute of Environmental Physics, University of Bremen. The data is derived using the ARTIST Sea Ice (ASI) algorithm developed by Spreen et al. (2008). The ASI retrieval method utilizes microwave radiometer data from the Advanced Microwave Scanning Radiometer for EOS instrument on the Aqua satellite and the Advanced Microwave Scanning Radiometer 2 instrument on the GCOM-W1 satellite. Both datasets were reprocessed in 2018 with the same parameters. To present the $20\%$ SIC, a seasonal average of the SIC is calculated for the years 2007 to 2010 and then gridded onto the $3.75°$x$1.9°$ grid using bilinear interpolation.

## 2.2 ERA5 Reanalysis

The ERA5 reanalysis data employs 4D-Var data assimilation of the ECMWF Integrated Forecast System. ERA5 incorporates an improved stratiform cloud and precipitation scheme, enhancing the representation of mixed-phase clouds compared to ERA-Interim (Hersbach et al., 2020) and includes prognostic variables for both rain and snow (Forbes et al., 2011; Forbes and Tompkins, 2011; Forbes and Ahlgrimm, 2014). The meteorological values are output at a resolution of $0.25$x$0.25°$.

In order to detect the presence of sLCCs in ERA5 data, specific criteria involving the T2m threshold and LWP threshold must be met (see Section 2.4). However, it is worth noting that ERA5 does not readily provide LWP values. Therefore, we calculated LWP using the total column cloud liquid water (tclw) and the total column rainwater (tcrw), expressed in $\mathrm{kgm}^{-2}$

(Table 1). We incorporate tcrw with the T2m threshold below $0°C$, as this threshold is used to exclude any rainwater below the melting layer. By using daily mean values of tclw and tcrw and applying the daily man temperature threshold, we can analyze the role of supercooled liquid water within clouds and the contribution of liquid water to the snowfall precipitation process in ERA5.

To calculate the $f_{snow}$, we rely on the mean snowfall rate variable (msr, Table 1). The mean snowfall rate, expressed in units of $kgm^{-2}s^{-1}$, provides a combined measurement of both large-scale and convective snowfall. It is calculated as an average over 24 hours. Here we use the ERA5 CDS tool to average all of the output variables into daily mean values based on hourly data (Cucchi et al., 2021; C3S, 2020-11-12). It is important to note that 2t, tclw, and tcrw (Table 1) are reported as instantaneous values every hour, meaning that when they are averaged to obtain daily mean values, they may not accurately represent the

actual daily mean values. Nevertheless, a non-zero model daily mean LWP means there was an sLCC in the grid box at some point during the day. In contrast, the mean snowfall rate is reported as a temporal average and accurately represents the daily mean snowfall rate.

## 2.3    CMIP6 Models

Our inter-model comparison relies on data from CMIP6, which involves simulations from various modeling centers. Specif-

ically, we focus on the *historical* simulations (1850-near present, Eyring et al., 2016), selecting ten models with daily mean outputs from different ensemble members as outlined in Table A1. In most of the figures shown in this study, we present the CMIP6 multi-model mean.

In our analysis, we consider the following variables (Table 1): mass mixing ratio of cloud liquid water (clw), encompassing liquid water in both large-scale and convective clouds provided at each model level in units of $kgkg^{-1}$. Precipitating hy-

drometeors within the clw variable are included only if they affect the model's radiative transfer calculations. Near-surface air temperature (tas), reflecting the temperature at a height of 2 meters above the surface, measured in Kelvin; and snowfall flux (prsn), representing precipitation in all forms of frozen water, expressed in $kgm^{-2}s^{-1}$.

As none of the models listed in Table A1 directly provide LWP as a daily output variable, we derive LWP based on clw. To calculate the LWP, we interpolate the CMIP6 hybrid-sigma pressure to isobaric pressure (Sec. A). The UKESM1-0-LL

and HadGEM3-GC31-LL models provide values on orographic vertical coordinates. Consequently, for these models, we use the CMIP6 variable phalf (Table 1) instead of calculating isobaric pressure levels. Subsequently, we utilize the hydrostatic equation to calculate the liquid water content for each vertical grid box. The liquid water content is then summed up over the vertical to obtain the LWP for each horizontal grid box (see Section A).

CMIP6 data is gathered from 2006 to 2009 as these years represent the closest available overlapping time range for all

CMIP6 models (some models ended the simulations by December 2009, marked with an asterisk in Table A1). The slight mismatch in the time range is of limited relevance, as CMIP6 model simulations are not designed to reproduce the exact temporal evolution of past weather but instead generate their own internal variability (e.g., ENSO cycles), so they should only be viewed as generally representative for the time period in question.

**Table 2.** Calculation of $f_{sLCC}$ and $f_{snow}$ and overview of applied thresholds. The first rows correspond to CMIP6/ERA5 metrics, while rows denoted with 'CC' describe CloudSat-CALIPSO metrics.

| Variable Name | Description | Thresholds and equations |
|---|---|---|
| N_all | Number of all days | |
| | CC: Number of all observations | |
| N_sLCC | Number of days with sLCCs | $T2m \leq 0°C$, $LWP \geq 5gm^{-2}$ |
| | CC: Number of observations with sLCCs | $T2m \leq 0°C$, phase flag liquid/mixed |
| N_sLCC_sf | Number of days with snowing sLCCs | $sf \geq 0.01kgm^{-2}h^{-1}$ |
| | CC: Number of observations with snowing sLCCs | $sf \geq 0.01kgm^{-2}h^{-1}$ |
| $f_{sLCC}$ (%) | Frequency of sLCCs | N_sLCC / N_all |
| $f_{snow}$ (%) | Frequency of snowing sLCCs | N_sLCC_sf / N_sLCC |

## 2.4 Calculation of $f_{sLCC}$ and $f_{snow}$

The instrument sensitivities of the CPR and CALIOP, as well as their different temporal and spatial sampling in contrast to the gridded ERA5 and CMIP6 data, require different metrics that ensure a like-for-like comparison to the extent possible. In the following, we describe the calculation of these metrics ($f_{sLCC}$ and $f_{snow}$) and call attention to differences between the satellite observations and ERA5 or CMIP6. For ERA5 and for CMIP6 models, we remap all models to the coarsest resolution of $3.75° \times 1.9°$, which corresponds to the grid of IPSL-CM5A2-INCA (Table A1) with a nominal resolution of approximately

500km. We use area-weighted averages for both the NH and SH to calculate spatial means.

### 2.4.1 Frequency of sLCC

We defined the occurrence of a sLCC in a cloudy CloudSat-CALIPSO profile when the lidar detected any liquid in the lowermost cloud layer (phase flag *liquid* or *mixed*), and the surface temperature was below freezing ($Temperature\_2m \leq 0°C$). The temperature threshold was applied to ensure that the clouds were supercooled. In ERA5 and CMIP6, we applied the same

temperature threshold and defined an sLCC when the LWP is above $5gm^{-2}$. This is in accordance with McIlhattan et al. (2017) and based on their sensitivity estimate when comparing CloudSat with ground-based microwave radiometer observations at Summit, Greenland. This definition of sLCCs is likely a very conservative estimate of the actual frequency of clouds containing supercooled liquid in the atmosphere because we require temperatures below freezing at the surface, while, in reality, there could still be supercooled liquid even if the surface is above freezing. However, as we are investigating the frequency of

snowing sLCCs in the second part of this study, we want to ensure that any precipitation from the sLCCs is primarily in the form of snow.

In ERA5 and CMIP6, we calculate $f_{sLCC}$ as the number of days with sLCCs present (N_sLCC) divided by the total number of days (N_all) over a specific time period (see Table 2). However, we cannot utilize daily statistics on a global scale from CloudSat-CALIPSO due to their transect-based sampling in combination with a revisit time of 16 days and, consequently, insufficient horizontal coverage (e.g., Kotarba, 2022; von Lerber et al., 2022). Thus, after aggregating the CloudSat-CALIPSO profiles to the same grid as ERA5 and CMIP6, we calculate $f_{sLCC}$ from satellite by dividing the number of profiles in that month with an sLCC by the total number of profiles (see Table 2).

### 2.4.2 Frequency of Snowfall in sLCCs

For the occurrence of snowfall from sLCCs, we define a snowfall threshold of $0.01 \mathrm{kgm}^{-2}\mathrm{h}^{-1}$. Again, this is based on McIllhattan et al. (2017) and aims to mitigate biases due to instrument sensitivities. The $f_{snow}$ in ERA5 and CMIP6 is then calculated as the number of days with snowfall (N_sLCC_sf) divided by the total number of days with sLCCs present (N_sLCC, see Table 2). The ERA5 and CMIP6 snowfall rates and T2m are given as daily mean values. It is important to note that as $f_{sLCC}$ is calculated using daily mean values, the simulated precipitation can, in principle, occur as rain (or supercooled rain) depending on the temperature at the time of the precipitation. However, as no additional information is available, we here assume that all precipitation from sLCCs is in the form of snow. To ensure comparability among the three datasets (CloudSat-CALIPSO, ERA5, and CMIP6), we perform a unit transformation, as outlined in Section B.

To account for the differences in sampling in space and time, we calculate $f_{snow}$ from CloudSat-CALIPSO as the number of observations with snowing sLCCs divided by the number of observations with sLCCs (Table 2).

## 3  Results

The following subsections examine and intercompare $f_{sLCC}$ and $f_{snow}$, respectively, from the three data sets examined (CloudSat-CALIPSO, ERA5 and CMIP6 models).

### 3.1  Frequency of sLCCs

Fig. 1 a-d presents the seasonal variability of $f_{sLCC}$ observed by CloudSat-CALIPSO in the mid-to-high latitudes of the NH. Evident from the figure is a clear seasonal progression in the spatial distribution of $f_{sLCC}$. In summer (JJA), $f_{sLCC}$ reaches its minimum spatial average, with non-negligible values primarily within the Arctic basin. In NH autumn (SON), $f_{sLCC}$ values increase significantly in the Arctic basin but also spread to land areas pole-ward of about $60°$N. In boreal winter (DJF), maximum values shift away from the Arctic basin and are instead found over the ocean close to the average seasonal sea ice edge (Fig. 1 a, red line).

The Greenland ice sheet (GRIS) is a region of particular interest for this study, as sLCCs have been proposed to accelerate GRIS surface melt (Bennartz et al., 2013; Hofer et al., 2019), while snowfall represents the primary ice sheet growth mechanism. The GRIS, which is characterized by its high elevation and snow-covered surface, experiences numerous sLCCs during boreal summer and autumn due to sufficient moisture and temperatures below $0°$C in the CloudSat-CALIPSO observations

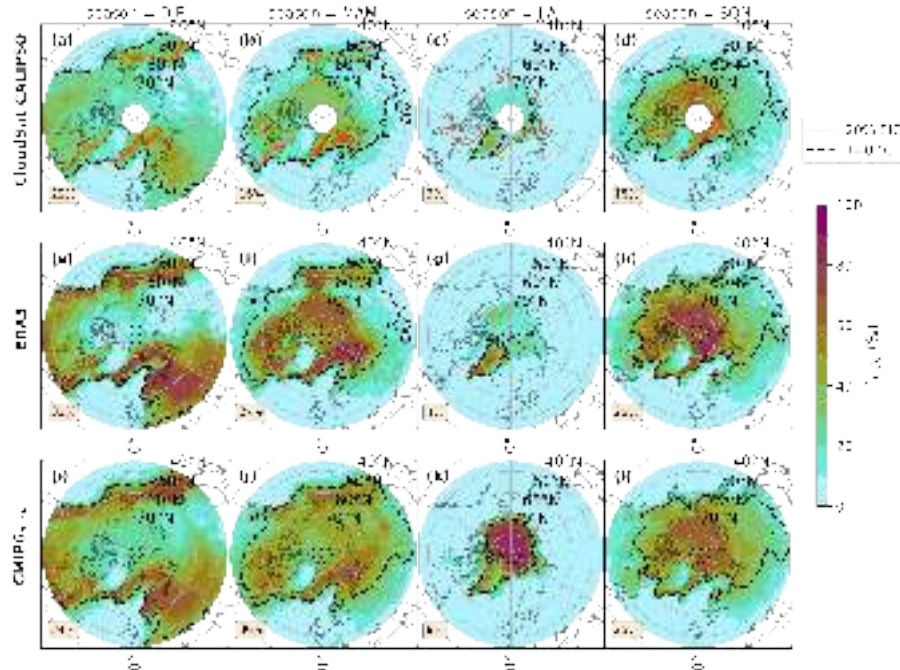

**Figure 1.** The NH mid-to-high latitude seasonal averages of $f_{sLCC}$. The first row (a-d) for CloudSat-CALIPSO, the second row (e-h) displays ERA5 data, and the third row (i-l) shows the CMIP6 model mean. Each map includes an area-weighted average for the study area (lower left corner). These averages are calculated for areas where CloudSat-CALIPSO have valid observations (between $45°N - 82°N$) and exclude the dotted area (in e-l). The black dashed line represents the seasonal mean 2m temperature $0°C$ isotherm for each individual product. The red line (in a-d) shows the average sea ice edge of $20\%$ sea ice concentration (SIC) between 2007 and 2010, for the given season.

(Fig. 1 c, d). However, during boreal winter and spring (MAM), the $f_{sLCC}$ is low ($< 15\%$) as the region becomes too dry and cold (Shupe et al., 2013) to support the formation of sLCCs. The seasonal cycle of $f_{sLCC}$ as observed by CloudSat-CALIPSO
is largely what would be expected from the combined seasonal influence of atmospheric circulation, moisture availability, and temperature conditions in the mid-to-high-latitudes in the NH.

Comparing the CloudSat-CALIPSO observations with ERA5 and the CMIP6 model mean, we generally observe that ERA5 and the CMIP6 models show a similar seasonal progression of $f_{sLCC}$ to that seen in the satellite observations. However, both ERA5 and the CMIP6 mean overestimate $f_{sLCC}$ to various extents during all seasons. The most significant discrepancy
is observed during boreal spring, with area-weighted averaged differences between CloudSat-CALIPSO and ERA5, and the CMIP6 model mean of $-11\%$ and $-14\%$, respectively (Fig. C1 f, j). Nevertheless, the spatial extent of areas with $f_{sLCC}$ values larger than $0\%$ in CloudSat-CALIPSO is replicated well in ERA5 and the CMIP6 model mean (Fig. 1). Although ERA5 and the CMIP6 models have remarkably similar spatial patterns of $f_{sLCC}$s, the CMIP6 model mean generally exhibits slightly larger area-weighted biases in the NH ($6\% - 14\%$, Fig. C1 i-l) than ERA5 ($1\% - 11\%$, Fig. C1 e-h) when compared
to CloudSat-CALIPSO.

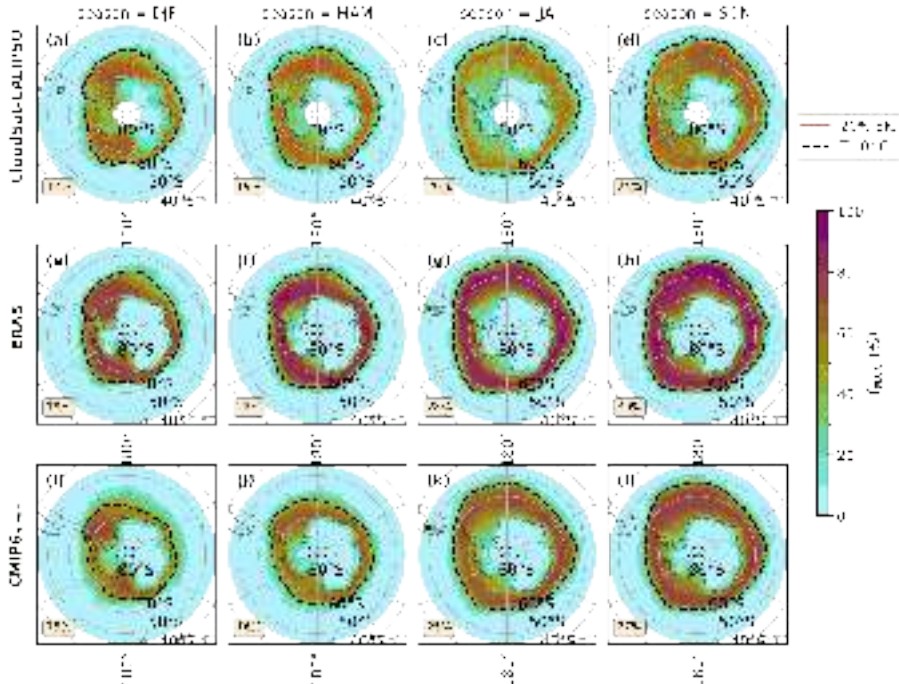

**Figure 2.** The SH mid-to-high latitude seasonal averages of $f_{sLCC}$. The first row (a-d) for CloudSat-CALIPSO, the second row (e-h) displays ERA5 data, and the third row (i-l) shows the CMIP6 model mean. Each map includes an area-weighted average for the study area (lower left corner). These averages are calculated for areas where CloudSat-CALIPSO have valid observations (between $45°\text{S} - 82°\text{S}$) and exclude the dotted area (in e-l). The black dashed line represents the seasonal mean 2m temperature $0°\text{C}$ isotherm for each individual product. The red line (in a-d) shows the average sea ice edge of $20\%$ sea ice concentration (SIC) between 2007 and 2010, for the given season.

The fact that these discrepancies are about equally present in both ERA5 and CMIP6 models provides valuable insights into their potential causes. They are most likely linked to the microphysical parametrizations of cloud processes that govern cloud phase in ERA5 and CMIP6 models. It is reasonable to assume that the temperature in the ECMWF-AUX product used in CloudSat-CALIPSO is quite similar to the ERA5 daily mean as indicated by the seasonal mean $0°\text{C}$ isotherm in Figs. 1 and 2.

However, ERA5 shows a slight variation in the $0°\text{C}$ isotherm line over Central Europe during DJF compared to ECMWF-AUX (Fig. 1 e). Furthermore, a comparison of the 2m temperature between ECMWF-AUX and ERA5 shows a latitudinal average difference of $0.24\text{K} \pm 0.22\text{K}$ (Fig. D1). Atmospheric circulation and overall cloud cover should be well constrained by the observations used in the ERA5 reanalysis. However, the cloud phase is not as evident in the identified biases. This finding will be discussed in greater depth in Section 5.1.

Unlike the NH mid-to-high latitudes, the SH does not experience significant seasonal variations in $f_{sLCC}$, according to the CloudSat-CALIPSO observations (Fig. 2 a-d). The $f_{sLCC}$ remains relatively constant across all seasons and is the highest in a band bounded by the Antarctic continent in the South and approximately the $60°$ parallel in the North, with values between $17\% - 24\%$. However, the area-weighted averages of $f_{sLCC}$ in the SH are generally higher than in the NH. The relatively

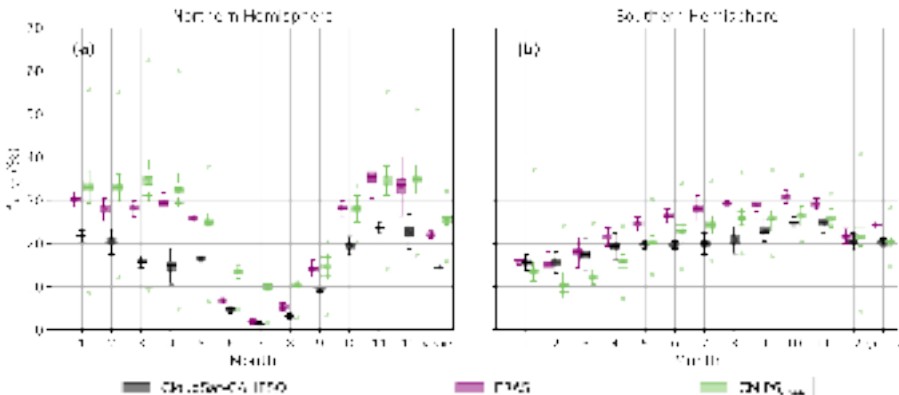

**Figure 3.** The annual cycle of monthly $f_{sLCC}$ for (a) NH (45°N-82°N), and (b) SH (45°S and 82°S). Shown are area-weighted averages for CloudSat-CALIPSO (black), ERA5 (pink), and the CMIP6 model mean (green). Each box represents the interquartile range (IQR) from the 25th to the 75th percentile. The whiskers extend to the minimum and maximum values, defined as the 25th percentile minus 1.5 times the IQR and the 75th percentile plus 1.5 times the IQR, respectively. Any points falling outside these whiskers are considered outliers and are marked with crosses. The green dots represent the minimum and maximum CMIP6 model values over all years.

constant $f_{sLCC}$ indicates that a more persistent cloud regime exists in the SH, with extensive sLCC cover present all year
round. This is consistent with previous literature reporting that the Southern Ocean is the region of the world with the most
extensive mixed-phase cloud cover (Matus and L'Ecuyer, 2017).

Like the GRIS, the Antarctic ice sheet has a low $f_{sLCC}$ with the lowest frequency in austral winter (JJA, $\leq 15\%$). With its
high elevations, the East Antarctic ice sheet exhibits a lower $f_{sLCC}$ than the flatter West Antarctic Ice Sheet during austral
summer (DJF), autumn (MAM), and spring (SON). Just like the NH, the SH shows a reduced area of $f_{sLCC}$ in austral summer
(Fig. 2 c), but with a larger extent towards lower latitudes (NH: $> 70°$N, SH: $> 60°$S). In all seasons, $f_{sLCC}$ are observed
primarily southward of the Antarctic Circumpolar Current, which may be due to the presence of warm water that results in
surface temperatures above the sLCC threshold (Fig. 2 a-d).

In both ERA5 and the CMIP6 model mean, there is a prominent gradient of sLCCs between the Southern Ocean and the
Antarctic ice sheet with $f_{sLCC}$ of up to $100\%$ over the Southern Ocean and $< 15\%$ over the Antarctic ice sheet. For the SH
spatial mean, ERA5 and CMIP6 model mean moderately overestimate $f_{sLCC}$ ($1\% - 8\%$), with the CMIP6 model mean, in
general, performing slightly better than ERA5 in comparison to CloudSat-CALIPSO (Fig. C2).

The biases in ERA5 and the CMIP6 models regarding $f_{sLCC}$, relative to CloudSat-CALIPSO, are further demonstrated
through the area-weighted averages for each month in both hemispheres (Fig. 3). For the area-weighted averages, we consider
only the locations where CloudSat-CALIPSO provided valid data (values between $|45° - 82°|$). CMIP6 models generate a
large spread of $f_{sLCC}$, with both CloudSat-CALIPSO and ERA5 falling within the range of the model maximum and minimum
values (green dots in Fig. 3). Again it is evident that ERA5 and the CMIP6 model mean have higher $f_{sLCC}$ than CloudSat-
CALIPSO in the NH (Fig. 3 a). The SH area-weighted averages show a better agreement between CloudSat-CALIPSO, ERA5,

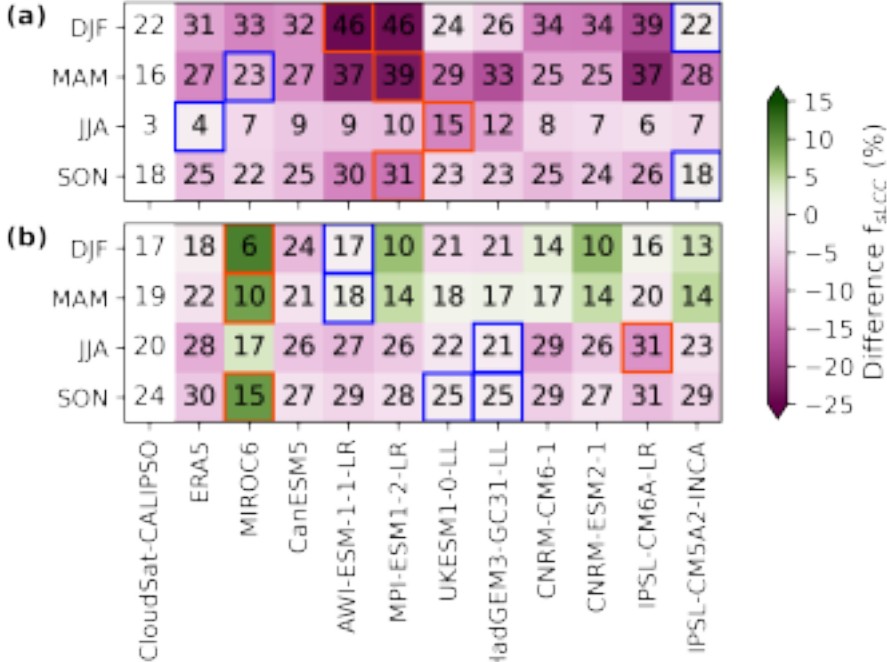

**Figure 4.** Magnitude of seasonal area-weighted averages (between $|45° − 82°|$) of $f_{sLCC}$ for CloudSat-CALIPSO, ERA5, CMIP6 models (numbers). The heatmap colors correspond to the differences of area-weighted averages of $f_{sLCC}$ between CloudSat-CALIPSO and ERA5 and CMIP6 models. Green (pink) values indicate underestimation (overestimation) of the individual model with respect to CloudSat-CALIPSO. (a) for the mid-to-high latitude NH, and (b) for the mid-to-high latitude SH. Per season, the smallest (largest) seasonal and spatial differences are outlined with blue (red) lines.

and CMIP6 model mean than in the NH (Fig. 3 b). The CMIP6 model mean underestimates $f_{sLCC}$ in comparison to CloudSat-CALIPSO during austral summer and autumn, but overestimates during winter and spring, while ERA5 overestimates $f_{sLCC}$
to various degrees all year round in the SH.

Finally, to assess the model performance relative to CloudSat-CALIPSO for ERA5 and the individual CMIP6 models for $f_{sLCC}$, we analyze the difference in seasonal and spatial averages for each of the 11 models (Fig. 4) for the NH and SH mid-to-high latitudes. The most significant discrepancy is seen in boreal winter and spring for AWI-ESM-1-1-LR, MPI-ESM1-2-LR, and IPSL-CM6A-LR with an negative difference $> 15\%$ (Fig. 4 a). In the NH, MPI-ESM1-2-LR is generally the model with
the larger overestimation ($> 15\%$), despite in boreal summer with a difference of $< 10\%$. IPSL-CM5A2-INCA has a low difference, $< 5\%$ in boreal summer, autumn, and winter. As previously highlighted, the difference values for $f_{sLCC}$ are lower in the SH (Fig. 4 b), and none of the models stand out in terms of overall performance in either direction.

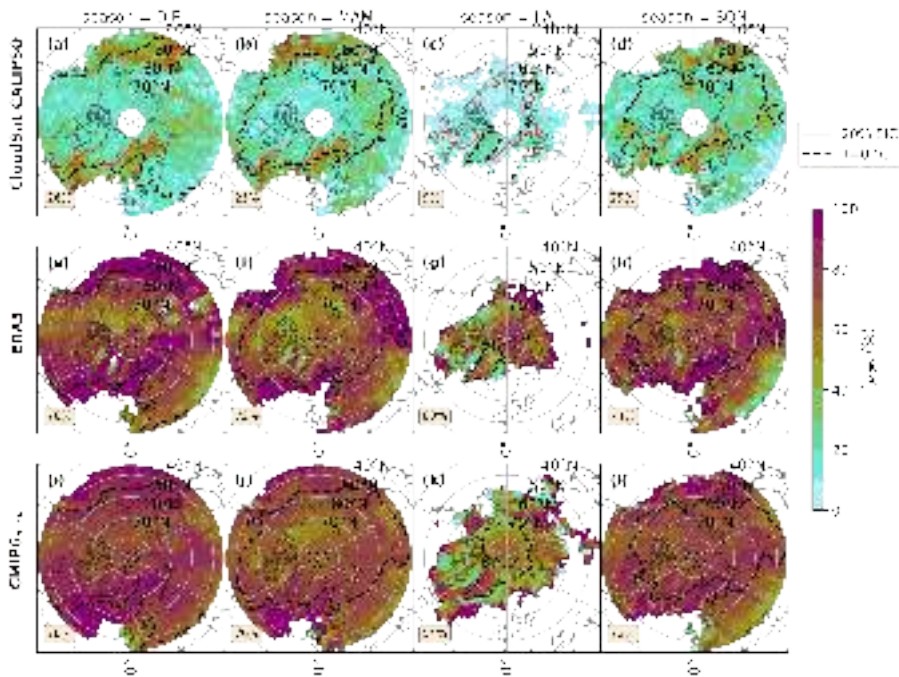

**Figure 5.** The figure presents the seasonal averages of $f_{snow}$ in the NH mid-to-high latitudes. The layout and area-weighted averages are calculated the same as those shown in Figure 1.

## 3.2 Frequency of Snowfall from sLCC

To assess the ability of ERA5 reanalysis and the CMIP6 models to represent snowfall processes, we compare the $f_{snow}$ between
them and the CloudSat-CALIPSO observations. Based on CloudSat-CALIPSO observations, NH sLCCs frequently produce snowfall, as can be seen by the non-zero values of $f_{snow}$ in Fig. 5 a-d. However, during boreal summer, the majority of sLCC rarely produce snowfall, as shown by the relative increase in the areal extent of zero $f_{snow}$ values (Fig. 5 c). Notably, during boreal winter, spring, and autumn, the seasonal average $f_{snow}$ is highest over open ocean regions, such as the Greenland, Norwegian, and Barents Sea ($30\% - 65\%$), the Labrador Sea ($40\% - 80\%$), and the Bering Sea ($40\% - 100\%$, Fig. 5 a, b, d).
This is likely due to plentiful available moisture for snowfall to occur and possibly linked to Cold Air Outbreaks (CAOs, Young et al., 2017). CAOs arise when cold, dry air flows over warmer, moist sea surfaces. CAOs occur frequently in the North Atlantic in boreal winter and spring, and the clouds associated with them are predominantly sLCCs (Fletcher et al., 2016; Papritz and Sodemann, 2018; Geerts et al., 2022; Mateling et al., 2023). While the influence of CAOs is also visible in the $f_{snow}$ patterns across various seasons, with the exception of boreal summer, the $f_{snow}$ values over land areas tend to be lower. Although it
is possible that the higher minimum height bin of the radar used over land ($\sim 1000$m) may lead to the 2C-SNOW-PROFILE product missing some of the precipitation over land, the variability in the $f_{snow}$ values in regions where similar cloud types are expected suggests that this land-sea contrast is likely a representative feature. This discrepancy can be attributed to the

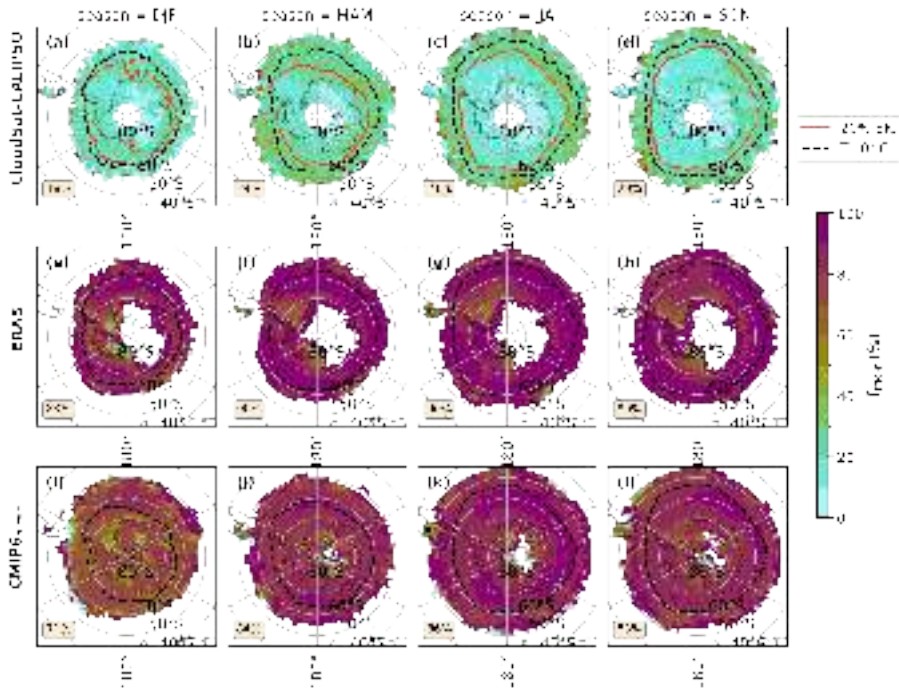

**Figure 6.** The figure presents the seasonal averages of the $f_{snow}$ in the SH study region. The layout and area-weighted averages are calculated the same as those shown in Figure 1.

limited moisture content and shallower boundary layer over land during winter, resulting in lower $f_{snow}$ compared to ocean areas where CAOs provide deeper, moisture-rich clouds. Once clouds embedded in CAOs reach a certain distance from the sea

ice, they frequently produce precipitation (e.g., Abel et al., 2017). In contrast, during the winter months, Northern Europe is often covered by non-precipitating, supercooled stratus clouds (Cesana et al., 2012; McIlhattan et al., 2017). The models also show a lower $f_{snow}$ over land, supporting the idea that the observed land-sea contrast is a genuine characteristic rather than an artifact of measurement techniques.

In contrast to the CloudSat-CALIPSO observations, both ERA5 and the CMIP6 model mean have much higher $f_{snow}$ values

($> 60\%$), which are evenly distributed over the NH mid-to-high latitudes, without a significant seasonal variability. In regions where all three datasets have sLCCs, ERA5 and the CMIP6 model mean overestimate the $f_{snow}$ by $\sim 50\%$ (Fig. C3). This indicates that ERA5 and the CMIP6 models produce snowfall much more frequently from sLCC than observed. It is important to note that since $f_{snow}$ accounts for the number of sLCCs, ERA5 and the CMIP6 models are too efficient in producing snowfall regardless of the number of sLCCs that are simulated/assessed.

Regarding the SH mid-to-high latitudes, the CloudSat-CALIPSO observations show that there is no significant seasonal variation in $f_{snow}$, with spatially averaged values of $\sim 20\%$ for all seasons (Fig. 6 a-d). However, when comparing the results from ERA5 and the CMIP6 model mean with observations from CloudSat-CALIPSO, we find that similar to the NH, the simulated

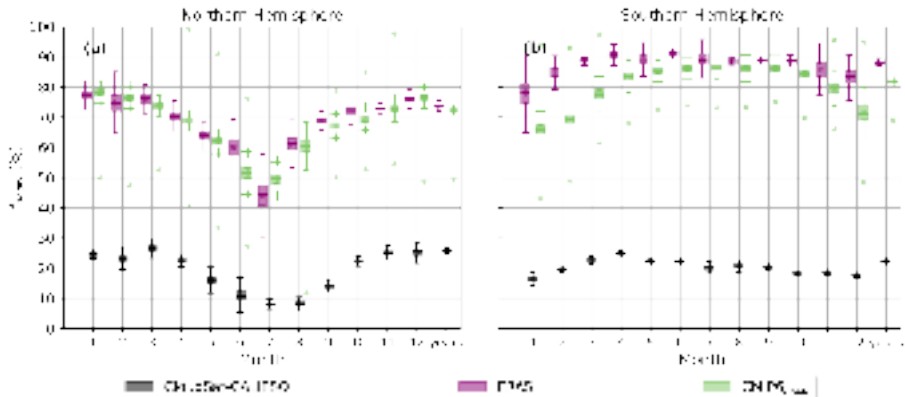

**Figure 7.** Annual cycle of $f_{snow}$ for (a) the NH, defined here as the region between $45°N$ and $82°N$ and (b) SH, defined here as the region between $45°S$ and $82°S$. Colors and boxplots are identical to Figure 3.

$f_{snow}$ values are much higher than those observed (Fig. 6). When considering only regions where CloudSat-CALIPSO observes $f_{snow}$ ($45°S - 82°S$), the area-weighted seasonal average of $f_{snow}$ is approximately $60\% - 70\%$ higher in ERA5 and the CMIP6 model mean (Fig. C4). The high simulated values of $f_{snow}$ in both the NH and SH indicate that ERA5 and CMIP6 models produce snowfall far too frequently from sLCCs compared to the satellite observations. In fact, when sLCCs are present, snow is produced practically all of the time (between $70\%$ and $90\%$). This bears resemblance to a well-established bias in ESMs, namely the persistent "perpetual drizzle" problem for warm liquid clouds (Mülmenstädt et al., 2020; Lavers et al., 2021).

To investigate the annual cycle of $f_{snow}$ and the interannual variation between 2007 and 2010, we present the area-averaged data based on grid boxes where CloudSat-CALIPSO is capable of making observations ($|45° - 82°|$, Fig. 7). CloudSat-CALIPSO displays a relatively clear seasonal cycle in $f_{snow}$ in the NH, but not in the SH. The difference in the $f_{snow}$ between boreal summer months and the rest of the year in the NH is $15\%$, while the SH value is constant around $20\%$ throughout the year. In contrast, ERA5 and the CMIP6 model mean display a more significant seasonal cycle, with a difference of $25\%$ between boreal summer and winter months (Fig. 7 a). Also, in the SH, there is more of a seasonal cycle evident for ERA5 and the CMIP6 model mean. Also worth noting is that ERA5 shows more pronounced interannual variability in both hemispheres of $f_{snow}$ for a given month, meaning that the year-to-year fluctuations are larger than for CloudSat-CALIPSO and the CMIP6 model mean.

The comparison of magnitude and model differences for $f_{snow}$ in Fig. 8 indicates how well the individual ESMs represent the frequency of occurrence for snow, and if the same models that perform well for $f_{sLCC}$ (Fig. 4) do so for surface snowfall as well. Among the models, AWI-ESM-1-1-LR and MPI-ESM1-2-LR come closest to matching CloudSat-CALIPSO observations, but their $f_{snow}$ are still much too high (NH: $\sim 30\%$, SH: $\sim 45\%$, Fig. 8). At the same time, IPSL-CM5A2-INCA deviates the most for $f_{snow}$ in the NH and SH, having a negative difference of $\sim 65\%$ and $\sim 75\%$, respectively. Interestingly, it appears that some models that perform well for $f_{sLCC}$ perform poorly for $f_{snow}$. This could be an indication that models that are able to simulate

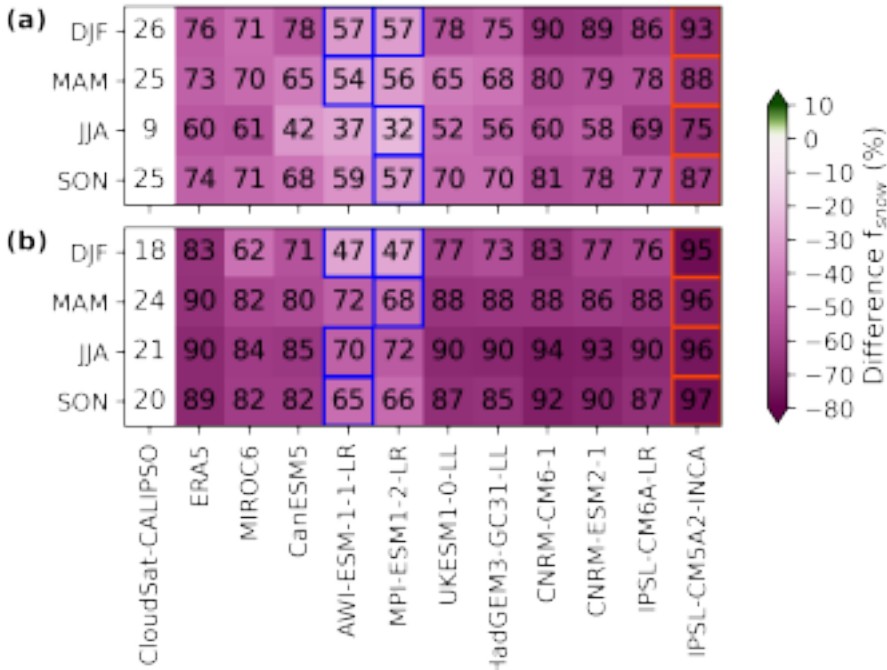

**Figure 8.** Magnitude of seasonal area-weighted averages (between $|45° − 82°|$) of $f_{snow}$ for CloudSat-CALIPSO, ERA5, CMIP6 models (numbers). The heatmap colors correspond to the differences of area-weighted averages of $f_{snow}$ between CloudSat-CALIPSO and ERA5 and CMIP6 models. Green (pink) values indicate underestimation (overestimation) of the individual model with respect to CloudSat-CALIPSO. (a) for the mid-to-high latitude NH, and (b) for the mid-to-high latitude SH. Per season, the smallest (largest) seasonal and spatial differences are outlined with blue (red) lines.

the right frequency of occurrence of sLCCs can do so because they are converting cloud condensate to snow too readily. In
other words, models may be getting the right answer for the wrong reason.

## 4   Sensitivity tests

In the previous section, we examined the frequency of occurrence of sLCCs ($f_{sLCC}$) and the frequency of occurrence of surface snowfall ($f_{snow}$) from sLCCs in CloudSat-CALIPSO, ERA5, and CMIP6 data. We found that the reanalysis and ESMs overestimate sLCCs and the frequency of surface snowfall in comparison to CloudSat-CALIPSO. These biases in the CMIP6
mean have potentially significant implications for our ability to predict how sLCCs and snowfall might change with future warming. At the same time, it is important to note that although the CMIP6 multimodel mean has this overestimation, some of the individual ESMs are much closer to the observations (green dots in Fig. 3), suggesting that these members may have more representative changes in clouds and snowfall in the future. It is therefore important to ensure that these findings are robust, and not overly reliant on subjective decisions or limitations in the design of the comparison. In the following subsections, we

investigate whether the identified discrepancies between the observations on one hand and ERA5 and the CMIP6 models on the other can be explained by sampling biases or instrument sensitivity.

It is important to acknowledge that CloudSat-CALIPSO data come with inherent uncertainties, including those associated with snowfall retrievals (Stephens and Kummerow, 2007; Hiley et al., 2011; Schirmacher et al., 2023). CloudSat surface snowfall is subject to sampling biases and ground clutter issues and also relies on ECMWF temperature data to differentiate between snowfall and rainfall (Boisvert et al., 2020). Milani et al. (2018) found that applying adjustments and a temperature threshold to the CloudSat snowfall retrieval led to a decrease in the estimated occurrence of snowfall events by up to 30%, primarily in the ocean regions surrounding Antarctica. Although these adjustments did not have the same effects everywhere, this highlights the sensitivity of the CloudSat retrievals to the assumptions made within them.

In addition to the uncertainties in the assumptions used in retrievals, comparing satellite observations with ESMs is challenging due to several factors linked to the sampling bias of the CloudSat-CALIPSO mission. Numerous considerations come into play when deriving climate statistics from satellite transects (Kotarba, 2022). These encompass the narrow swath coverage, infrequent revisits, and latitude-dependent ground track density (Kotarba, 2022; von Lerber et al., 2022). Varying the domain size and time scales of CloudSat observations showed that data on smaller scales or shorter time periods can introduce significant uncertainties associated with cloud, aerosol, or atmospheric properties (Henderson et al., 2013). However, instantaneous errors (using satellite data on smaller scales or shorter periods) tend to cancel out over longer time periods (Henderson et al., 2013; von Lerber et al., 2022). Another limitation of space-based remote sensing observations is the uncertainty associated with the retrieval method employed to derive the snowfall rate from radar reflectivity measurements (Kulie and Bennartz, 2009; Milani et al., 2018). However, in our study, we examine frequencies of occurrence and only use the surface snowfall amount to classify if an sLCC is snowing ($\mathrm{sf} \geq 0.01\,\mathrm{kgm}^{-2}\mathrm{h}^{-1}$) or not. Such a binary variable should be less sensitive to the exact snowfall rate retrieval. Edel et al. (2020) presented a CloudSat snowfall climatology, the frequency of snow, and snowfall rates over the Arctic. They found that the distribution of snowfall rate does not always match the distribution of snowfall frequency in CloudSat. Nevertheless, future modeling studies should ideally use the Cloud Feedback Model Intercomparison Project Observation Simulator Package (COSP, Bodas-Salcedo et al., 2011) to make satellite observations and model datasets more comparable. At present, COSP output is not available for a sufficient number of models and at a high enough temporal resolution to be useful for the present study.

Nevertheless, these satellite observations serve as valuable tools for identifying potential biases in reanalysis and ESMs, as demonstrated by previous studies (e.g., McIlhattan et al., 2017; Milani et al., 2018; Daloz et al., 2020; Heymsfield et al., 2020; Boisvert et al., 2020). To test the robustness of the overall findings of this study to our comparison approach, we next want to show how different temporal resolutions of ERA5, as well as different LWP thresholds applied to the CMIP6 data, impact the results.

## 4.1 Sensitivity to ERA5 temporal resolution

In the comparison by McIlhattan et al. (2017), the analysis was performed on a 6-hourly instantaneous model output to define the occurrence of LCCs in CESM-LE. As we compare ERA5 and CMIP6 models to CloudSat-CALIPSO, we only have daily

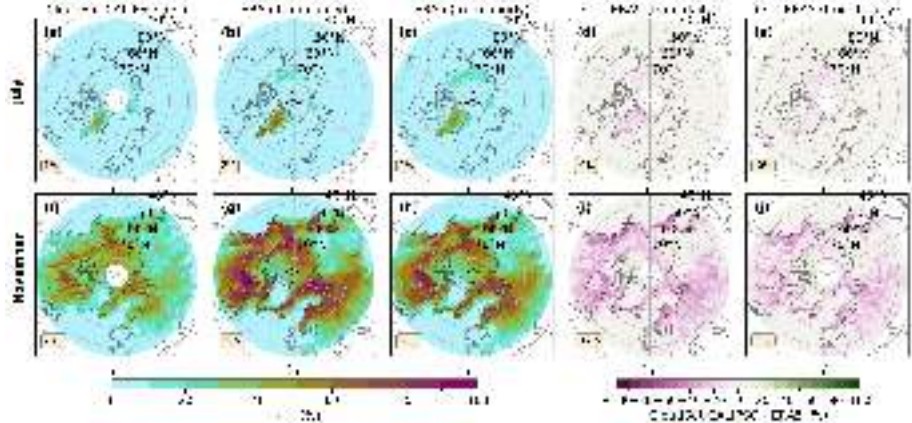

**Figure 9.** NH mid-to-high latitude monthly averages of $f_{sLCC}$ for July (a-e) and November (f-j) 2007 to 2010. The columns show the CloudSat-CALIPSO (CC) monthly means (a, f), ERA5 monthly means based on daily average values (b, g), and ERA5 monthly means based on hourly values (c, h). The fourth and fifth columns are the difference plots between CloudSat-CALIPSO and the ERA5 daily averages (d, i) and ERA5 hourly values (e, j), respectively. In the last two columns, green (pink) values indicate underestimation (overestimation) with respect to the reference used. Area-weighted averages for the study area ($45°\text{N} - 82°\text{N}$) are located in the lower-left corner of each map and exclude the dotted area (in b, c, g, h).

means of LWP available. Thus, instead of four instantaneous values per day for a given model grid box, we use only one
daily mean value of LWP. This way, we are comparing 30 daily values from ERA5 and CMIP6 models with thousands of
CloudSat-CALIPSO profiles within each grid box to calculate $f_{sLCC}$. Due to the 16-day repeat cycle of CloudSat-CALIPSO,
we cannot perform the satellite analysis on a daily temporal resolution at the given spatial resolution of the models. While
CMIP6 output is not available at a higher temporal resolution, for ERA5, we can actually test the sensitivity using an even
higher time resolution than McIlhattan et al. (2017). We take the hourly outputs from ERA5 from July and November 2007
to 2010 and create monthly mean values. July and November are chosen as these are the months with the lowest and highest
$f_{sLCC}$ in CloudSat-CALIPSO (Fig. 3 a). We find that the $f_{sLCC}$ values emerging from ERA5 data are not very sensitive to the
output frequency, as shown in Figure 9 (d-e, i-j). The area-weighted difference between CloudSat-CALIPSO and ERA5 hourly
values is negligible in July ($1\%$) and $\sim 15\%$ in the inner Arctic and reduces from $12\%$ overestimation to $7\%$ in November.
However, the conclusion that ERA5 overestimates the $f_{sLCC}$ does not change by changing to a higher time resolution (Fig. 9).
While the sensitivity analysis for ERA5 data indicates that changes in output frequency do not significantly affect the results,
the same conclusions cannot be directly extrapolated to CMIP6 models. However, based on our analysis, it is reasonable to
assume that similar results could be observed for individual ESMs within CMIP6, although this cannot be confirmed.

### 4.2    Sensitivity to LWP threshold for CMIP6 models

McIlhattan et al. (2017) conducted a similar analysis as in this study by examining the frequency of Arctic LCCs and $f_{snow}$.
They experimented with different LWP thresholds. Specifically, they used the $5\text{gm}^{-2}$ and $0.01\text{gm}^{-2}$ thresholds and based

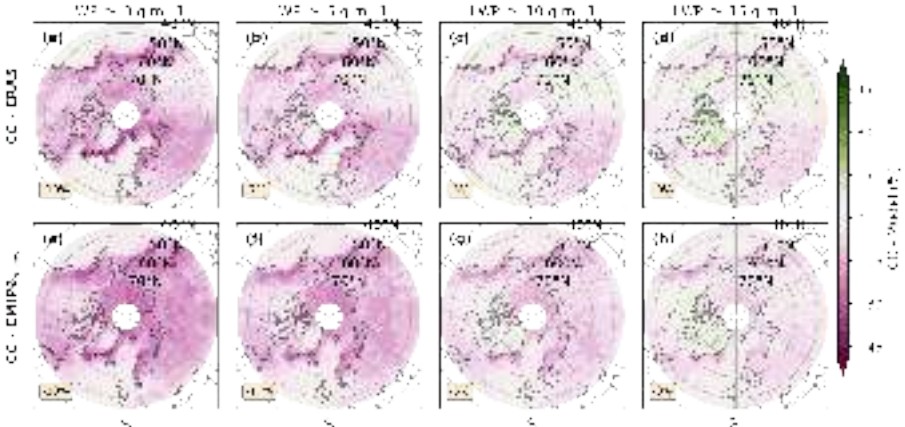

**Figure 10.** Annual average difference plots (CloudSat-CALIPSO (CC) minus model) for $f_{sLCC}$ in the NH mid-to-high latitudes. The first row represents ERA5 data (a-d), and the second row shows the CMIP6 model mean (e-h) difference. LWP thresholds applied of $3\text{gm}^{-2}$ (a, e), $5\text{gm}^{-2}$ (b, f), $10\text{gm}^{-2}$ (c, g), and $15\text{gm}^{-2}$ (d, h). The analysis in this study focuses on the LWP threshold of $5\text{gm}^{-2}$. The maps in each row are accompanied by area-weighted averages for the study area, located in the lower-left corner of each map. The averages are calculated for areas where CloudSat-CALIPSO observations have valid $f_{sLCC}$ values.

their LWP threshold of $5\text{gm}^{-2}$ on the approximate retrieval uncertainty of LWP from ground-based microwave radiometer observations. We performed sensitivity tests in which we varied the threshold between 3, 5, 10, and $15\text{gm}^{-2}$ (Fig. 10). While some of these values are unreasonably high, it is nevertheless useful to test how adjusting the LWP threshold value affects the identified bias between the observations and simulations. As expected, Figures 10 and E1 indicate that as the LWP threshold

increases, the overestimation of $f_{sLCC}$ decreases both in the NH and SH. However, unless we increase the LWP threshold in ERA5 and in the CMIP6 models to an unrealistically high value, the general findings and conclusions hold - ERA5 and the CMIP6 model mean still overestimate the occurrence of sLCCs in the study area in comparison to CloudSat-CALIPSO.

## 5 Discussion and Conclusion

We find that mid-to-high latitude sLCCs and snowfall are produced more frequently in ERA5 and CMIP6 than in CloudSat-

CALIPSO. While previous studies have focused on the underestimation of supercooled liquid fraction (SLF) in mixed-phase clouds (Komurcu et al., 2014; Cesana et al., 2015; Tan and Storelvmo, 2016; Kay et al., 2016; Bruno et al., 2021; Shaw et al., 2022), this research focuses on the frequency of occurrence of sLCCs. This difference in the cloud phase metric can lead to seemingly contradicting conclusions. We illustrate why our results do not necessarily contradict previous findings with the following example: If an ESM consistently predicts a low percentage of liquid in clouds, while satellite observations suggest

that liquid occurs only $50\%$ of the time but with a high liquid percentage when it does occur, it can create a discrepancy between the metric used here and other cloud phase metrics like SLF. Even though the model would then have an sLCC frequency of occurrence of $100\%$, the amount of supercooled liquid could still very well be underestimated. When combined

with previous studies finding that ESMs generally underestimate SLF, our findings suggest that models too frequently simulate genuinely mixed-phase clouds with both phases present at the same time and point, while observations more frequently reveal conditionally mixed conditions, in which ice and liquid are separated in time and/or space (Korolev and Milbrandt, 2022).

Previous studies not only use different metrics but also, in some cases, evaluate different generations of climate models, each with distinct cloud characteristics. For instance, many models included in CMIP5 were found to have too few sLCCs (Cesana et al., 2012, 2015; Tan and Storelvmo, 2016; Kay et al., 2016; Lenaerts et al., 2017; McIlhattan et al., 2017). This deficiency was addressed in several of the next-generation models used in CMIP6, resulting in an increase in the simulated SLF (Mülmenstädt et al., 2021). Although some CMIP6 models used in this study still over- or underestimate the traditional SLF metrics compared to CALIPSO observations, models such as CNRM-CM6-1 and CNRM-ESM2-1 are within one standard deviation of the satellite observations (Fig. F1). Studies focusing on high latitudes have shown that comparing individual CMIP5 models with CMIP6 models reveal improved cloud representation, more closely aligning with observations due to microphysical adjustments in the newer model versions (Lenaerts et al., 2020; McIlhattan et al., 2020). Indeed, McIlhattan et al. (2020) showed that when going from the CMIP5 to the CMIP6 version of CESM the frequency of LCC bias switched direction and overestimates in comparison to CloudSat-CALIPSO. Lenaerts et al. (2020) showed that cloud coverage increased and is slightly overestimated in the CMIP6 version of CESM, but still underestimates ice water path. Consequently, some of the differences between our findings and those from previous studies may be attributed to these improvements in the newer model generations.

## 5.1 Links between biases in $f_{sLCC}$ and $f_{snow}$

We find that ERA5 and CMIP6 overestimate the magnitude of $f_{sLCC}$ and $f_{snow}$ in both hemisphere's mid-to-high latitudes when compared to CloudSat-CALIPSO. In contrast, McIlhattan et al. (2017) showed that the CESM-LE underestimates the $f_{LCC}$ by $\sim 17\%$ and overestimates the $f_{snow}$ by $\sim 57\%$ in the Arctic. However, since we utilize a different metric (sLCC instead of LCC), there is no reason to expect the model biases to be identical. Nevertheless, in a further study by McIlhattan et al. (2020) it was shown that the LCC frequency in the newer CESM version is more aligned with the satellite observations except for in the summer months, where it overestimates the LCC frequency.

Similarly, we generally observe that ERA5 and the CMIP6 mean overestimate the $f_{sLCC}$ to various extents during all seasons. These overestimations are likely linked to the microphysical parametrizations of cloud processes that govern cloud phase. This finding aligns with McIlhattan et al. (2020), indicating that while newer model versions have advanced in representing LCC frequencies more accurately, they can still overestimate cloud occurrences, particularly in specific seasons. Precipitation in the new CESM version is more frequent but lighter overall compared to the previous version, which is similar to our findings indicating that the models's sLCCs produce continuous snowfall, analogous to the "perpetual drizzle" problem (Mülmenstädt et al., 2020; Lavers et al., 2021).

Furthermore, while McIlhattan et al. (2017, 2020) considered a single ESM, our study considers an ensemble of CMIP6 models. Nevertheless, the insights from McIlhattan et al. (2017, 2020) provide relevant context for our finding that ERA5 and the CMIP6 model mean produce sLCCs more frequently than observed in the NH and SH mid-to-high latitudes, especially over

the sea ice and land depending on the season (Figs. C1 and C2). In these regions, not only is the frequency of occurrence of sLCCs too high, but the sLCCs are too efficient at producing snowfall in ERA5 and CMIP6 models (Figs. 5 and 6). The latter finding is consistent with the findings of McIlhattan et al. (2017) that LCCs produce snow too frequently in the CESM-LE model.

McIlhattan et al. (2017) explained the overestimation of $f_{snow}$ from LCC by exploring one potential microphysical pathway for removing supercooled cloud liquid - the WBF process. An overactive WBF process and subsequent removal of cloud ice from the cloud through snow formation in ERA5 and CMIP6 models could also explain the large overestimation of $f_{sLCC}$ and $f_{snow}$ in ERA5 and the CMIP6 model mean. ERA5 and CMIP6 models may thus have more frequent but lighter snowfall events that lead to a higher $f_{snow}$ while maintaining the clouds longer, compared to the CloudSat-CALIPSO observations. However, for some ESMs, there is no explicit simulation of the WBF process, but rather a simple temperature-dependent partitioning of cloud condensate into liquid and ice. For such crude parameterization schemes the identified biases are inevitable, as clouds at a given mixed-phase temperature would always have some liquid present (hence a very high $f_{sLCC}$) and at the same time always have ice available that could be converted to snow (hence a very high $f_{snow}$).

The difference in simulated $f_{sLCC}$ and $f_{snow}$ compared to CloudSat-CALIPSO in the NH and SH could, in theory, result from differences in surface conditions and/or weather patterns, and thereby differences in moisture availability for the formation of sLCCs (McIlhattan et al., 2017). However, the fact that very similar biases are found in the reanalysis as in the CMIP6 models might suggest that surface conditions and weather patterns are not the main explanation for the identified biases. That being said, reanalysis products are known to be less reliable at high latitudes (Liu and Key, 2016). We therefore cannot rule out differences in weather patterns as a partial cause of the biases discussed above, but maintain that deficiencies in the model representation of cloud microphysics is the more likely culprit. The inclusion of both ERA5 and CMIP6 models in this analysis is thus beneficial as it helps highlight potential root causes of the identified biases, although it should be noted that individual CMIP6 models do not all show biases that match ERA5 (Figs. 4 and 8). As stated above, we cannot entirely rule out a modest contribution to the biases from circulation differences, even for the reanalysis. For example, according to Boisvert et al. (2020), all reanalysis products show lower precipitation amounts in the southern Ross Sea and Weddell Sea embayments near the ice shelves. They explain that this is due to persistent cold and dry katabatic wind blowing from the continent across the ice shelves and out over the sea ice, which is not beneficial for precipitation formation and not captured well by the reanalyses.

Under the assumption that the overall weather patterns and surface conditions are accurate in ERA5, the most plausible root cause of the biases in $f_{sLCC}$ and $f_{snow}$ are linked to the microphysical parametrizations in the reanalysis and ESMs (Kiszler et al., 2023).

Boisvert et al. (2020) suggests that the difference in temperature and temperature threshold among various reanalysis microphysical products may explain the latitudinal inequality in snowfall amount observed between CloudSat and the reanalysis products. But here, no strong latitudinal dependence is observed in Fig. 6 as we investigate $f_{snow}$ instead of snowfall amount. The same principle should apply to ERA5 and the CMIP6 models as they rely on microphysical parametrizations. For the most part, the $0°C$ isotherm shown in Figs. 1 and 2 supports our argument that the primary issue with the ERA5 and CMIP6 datasets lies not with the simulated temperature itself, but with the representation of cloud properties and microphysics. This

distinction highlights that the observed deviations in $f_{sLCC}$ and $f_{snow}$ are driven more by inaccuracies in cloud simulation than by temperature discrepancies. Exceptions occur over Central Europe during DJF between ECMWF-AUX, ERA5 and the CMIP6 ensemble mean (Figs. 1 a, e, i). However, more notably is the difference in the CMIP6 ensemble mean over the central

Arctic during summer (Fig. 1 k), where simulated temperatures appear to be too cold. In this specific case, the cloud bias could stem from a temperature bias, suggesting a potential link between temperature inaccuracies and cloud simulation in the CMIP6 ensemble mean for this region and season. Figs. 4 and 8 show some models performing better than others for $f_{sLCC}$ and $f_{snow}$ in comparison to CloudSat-CALIPSO. As discussed above, many models have a simple temperature-dependent cloud phase that would almost certainly cause them to overestimate the $f_{sLCC}$ and $f_{snow}$. A follow-up study will investigate the SLF by isotherm

for mixed-phase clouds of several CMIP6 models. Figure F1 shows that some of the CMIP6 models have a more variable SLF in time and space (e.g., AWI-ESM-1-1-LR, MPI-ESM1-2-LR), which might be an indication that they could have more sophisticated microphysics and could, therefore, perform better in terms of sLCC and surface snowfall occurrence. However, it is not immediately clear which models use a purely temperature-dependent cloud phase partitioning. From the interquartile range (IQR, Fig. F2) CNRM-CM6-1, CNRM-ESM2-1, and IPSL-CM6A-LR are assumed to have a simple temperature-dependent

cloud phase due to the smaller IQR of the temperature at various SLF. Interesting to note, these models with the simpler temperature-dependent schemes are the most poorly performing models for $f_{snow}$ (Fig. 8).

Imura and Michibata (2022) studied the effect of changing the microphysical parametrization scheme from the traditional diagnostic scheme used in MIROC6, participating in CMIP6, to a prognostic scheme. They found that the prognostic scheme improved cloud coverage and snowfall in the Arctic. In our study, MIROC6 is one of the models with the largest overestimation

of $f_{sLCC}$ in the SH and the model with the smallest difference in boreal spring (Fig. 4). Furthermore, MIROC6, with the diagnostic scheme used here, has a large negative difference in $f_{snow}$ ($45\% - 60\%$) independent of hemisphere and season (Fig. 8). However, the MIROC6 with the prognostic parametrization scheme in Imura and Michibata (2022) produced light snowfall too frequently, in agreement with the results presented here, despite the use of a different microphysical scheme.

ERA5 and the CMIP6 model mean are not able to reproduce $f_{snow}$ in comparison to CloudSat-CALIPSO. The overestimation

in ERA5 could be related to the global average wet bias of up to $0.27\mathrm{mmday}^{-1}$ in the ECMWF product used in ERA5 (Lavers et al., 2021). If the modeled sLCCs are practically always snowing, then the sLCCs should have a shorter lifetime and subsequently have a lower $f_{sLCC}$ than observations suggest. However, ERA5 and the CMIP6 model mean seem to have sLCCs that produce a little bit of snow all the time and, at the same time, maintain the sLCCs in the atmosphere. This bias may be the sLCC's counterpart to the "perpetual drizzle problem" that has been identified in ESMs for warm liquid clouds (Mülmenstädt

et al., 2020).

## 5.2 Implications for modeling and future projections

An overestimation of snowfall frequency can have significant effects on the Earth's radiative budget. The fact that the models seem to snow a little bit all the time instead of producing occasional heavy snowfall events, as well as periods with no snowfall, is therefore concerning. One implication is that there will be too frequent fresh snow deposited on top of the snow pack, ice

sheets, sea ice and land surfaces, with consequences for the surface albedo. With time, the aging snow surfaces darken due

to snow/ice metamorphism processes and the deposition of absorbing aerosols (e.g., Picard et al., 2012; Carlsen et al., 2017). Continuous light snowfall in the model could thus lead to a simulated overestimation of the surface brightness, which in turn could limit sea ice melt in ESMs. Indeed, only ESMs that simulate excessive Arctic warming are able to reproduce the observed sea ice loss in recent decades (Notz and Stroeve, 2016).

The overestimations of $f_{sLCC}$ and $f_{snow}$ could stem from how models handle cloud microphysical processes, particularly the WBF process and the partitioning of cloud condensates into the liquid and ice phase. Some identified biases in ESMs may be due to the specific parametrization schemes used in the models, which could rely on simple temperature-dependent cloud phase partitioning. For example, models with a temperature-dependent cloud phase could simulate a fixed liquid fraction at a given temperature within the mixed-phase range, making it impossible for them to simulate an all-ice or all-liquid cloud in that range. Consequently, these models could tend to under- or overestimate the occurrence of sLCCs.

        Interestingly, models with more sophisticated microphysics schemes (Figs. F1 and F2) do not necessarily perform better (Figs. 4 and 8). Larger biases in cloud phase can still occur, for example if these models do not accurately represent INPs. This underscores how generational advancements and implementations of observational constrains can significantly influence the comparison and interpretation of cloud and precipitation patterns.

Continuing work should be performed on the comparison between ESMs, reanalysis and observations at specific locations to better understand microphysical processes and their representation in ESMs and reanalysis data. Cloud microphysical scheme development should, therefore, focus on improving these processes in order to reduce cloud phase and surface snowfall biases. More sophisticated schemes should be explored, and ensemble studies with varied microphysical model parameterizations can provide insights into how different parameterizations affect the representation of sLCCs and snowfall in climate simulations.

Ground and aircraft observations in connection with field campaigns at specific locations should, therefore, be increased, especially to improve our understanding of cloud phase, liquid and ice water content, and cloud top temperatures in regions where sLCCs form. These field campaigns should also focus on the processes responsible for snowfall from these clouds.

        Finally, as shown in the present study, satellite observations are an essential validation tool for assessing the representations of clouds and precipitation in ESMs. To ensure that future model generations are as accurate as possible, continuous improvements to satellite retrievals are required to further refine our understanding of cloud and snowfall properties, especially in high-latitude regions where limitations in sun angle and ground-track are prevalent. The accurate representation of these cloud and snowfall processes in models, validated through enhanced satellite observations, is essential for reliable simulations of weather and climate. This accuracy is particularly important for making future predictions of the hydrological cycle, which are essential for understanding and mitigating the impacts of climate change.

*Code availability.* The code used to analyze the satellite, ERA5, and CMIP6 data and to produce the figures is available via https://github.com/franzihe/CloudSat_ERA5_CMIP6_analysis

*Data availability.* ERA5 meteorological parameters downloaded via the ERA5 CDS tool for daily statistics (C3S, 2020-11-12). CMIP6 model variables downloaded via ESGF-WCRP for 2006-2009 (https://esgf-node.llnl.gov/search/cmip6/). The standard CloudSat (Stephens et al., 2002) and CALIPSO (Winker et al., 2010) data products (version R05) used in this study (2B-CLDCLASS-LIDAR, 2C-SNOW-PROFILE, ECMWF-AUX) were downloaded from the CloudSat Data Processing Center's (at Cooperative Institute for Research in the Atmosphere, Colorado State University, Fort Collins) website (http://www.cloudsat.cira.colostate.edu). The sea ice concentration data is from the Institute of Environmental Physics (IUP), University of Bremen, based on the ARTIST Sea Ice (ASI) algorithm (Spreen et al., 2008). The daily data sets (Melsheimer and Spreen, 2019a, b, 2020a, b) were downloaded for the years 2007–2010 from the data publisher PANGAEA.

## Appendix A: CMIP6 models and LWP calculation

The daily averages of the total mass of liquid water (LWP) in a grid box are calculated using the hydrostatic equation in CMIP6 models, besides the models from the modeling institute MOHC. First, isobaric pressure levels are retrieved with the individual CMIP6 hybrid sigma model formula found in Table A1. $\mathrm{hyam}$ and $\mathrm{hybm}$ represent the vertical coordinate formula term along the dimension $\mathrm{Nlev}$. $\mathrm{p_{sfc}}(\mathrm{i,j})$ is the surface air pressure for each latitude (i) and longitude (j) grid box, and $\mathrm{p_0}(\mathrm{Nlev})$ the vertical coordinate formula term: reference pressure, both in Pa for the individual model. For CMIP6 models with orographic vertical coordinates (UKESM1-0-LL, HadGEM3-GC31-LL, Table A1), represent $\mathrm{bm}(\mathrm{Nlev})$ the vertical coordinate formula term and $\mathrm{orog}(\mathrm{i,j})$ the surface altitude in units of metre.

After retrieving the pressure coordinate on half isobaric pressure levels $(\mathrm{p}(\mathrm{i,j,Nlev} - (\mathrm{k} \pm {}^1\!/_2)))$ and $\mathrm{clw}(\mathrm{i,j,Nlev})$ on the full isobaric pressure levels (Nlev) in CMIP6 the $\mathrm{lwp}$ in each vertical grid box can be calculated with the hydrostatic equation.

$$\frac{\Delta p}{\Delta Z} = -\rho_{air}(i,j,Nlev) \cdot g$$

$\rho_{\mathrm{air}}(\mathrm{i,j,Nlev})$ is the density of the air mass at full isobaric pressure level. $\Delta Z$ depicts the height difference between the half isobaric pressure levels, and $\mathrm{g} = 9.81 \mathrm{ms}^{-2}$ is the gravity acceleration. After the AMS Glossary of Meteorology the $\mathrm{lwp}(\mathrm{i,j,Nlev})$ is defined as

$$lwp(i,j,Nlev) = \int\limits_{z=0}^{\infty} \rho_{air}(i,j,Nlev) \cdot clw(i,j,Nlev) \cdot dz = \int\limits_{0}^{p=p_0} \rho_{air}(i,j,Nlev) \cdot clw(i,j,Nlev) \cdot \left(-\frac{dp}{\rho_{air} \cdot g}\right)$$

$$lwp(i,j,Nlev) = \int\limits_{p=p(i,j,Nlev-(k-^1/_2))}^{p=p\left(i,j,Nlev-(k+^1/_2)\right)} \frac{clw(i,j,Nlev)}{g} \cdot dp$$

The $\mathrm{LWP}(\mathrm{i,j})$ can then be calculated by summing the individual $\mathrm{lwp}(\mathrm{i,j,Nlev})$ per vertical grid box from the surface to the top of the atmosphere. And it follows for the daily mean total liquid water path per column air in each pixel:

$$LWP(i,j) = \sum_{k=0}^{NLEV+1} lwp(i,j,Nlev) \tag{A1}$$

$$= -\frac{1}{g} \sum_{k=0}^{NLEV+1} clw(i,j,Nlev) \cdot \left[p\left(i,j,Nlev-(k-^1/_2)\right) - p\left(i,j,Nlev-(k+^1/_2)\right)\right] \tag{A2}$$

**Table A1.** This study uses the Coupled Model Intercomparison Phase 6 (CMIP6) models. Models marked with an asterisk (∗) have some *historical* simulations only until 2009. Isobaric levels are calculated with a formula given by the individual model. The three equations used are ($\bigotimes$) $p(i, j, Nlev) = hyam(Nlev) \cdot p_0 + hybm(Nlev) \cdot p_{sfc}(i,j)$, ($\bigcirc$) $p(i, j, Nlev) = hyam(Nlev) + hybm(Nlev) \cdot p_{sfc}(i,j)$, ($\triangle$) $z(i, j, Nlev) = am(Nlev) + bm(Nlev) \cdot orog(i,j)$. Further description of the ESMs is available on the ES-DOC interface (https://explore.es-doc.org/cmip6/models/ last accessed 25. Oct 2023).

| Institution | Model Name | Nom. Res. | Levels | Top level | Atmosphere | Variant | Reference |
|---|---|---|---|---|---|---|---|
| MIROC | $\bigotimes$ MIROC6 | 250km | 81 | 0.004hPa | CCSR AGCM | r1i1p1f1 | Tatebe et al. (2019) |
| CCCma | $\bigcirc$ CanESM5 | 500km | 49 | 1hPa | CanAM5 | r1i1p2f1 | Swart et al. (2019) |
| AWI | $\bigcirc$ AWI-ESM-1-1-LR | 250km | 47 | 80km | ECHAM6.3.04p1 | r1i1p1f1 | Ackermann et al. (2020) |
| MPI-M | $\bigcirc$ MPI-ESM1-2-LR | 250km | 47 | 0.01hPa | ECHAM6.3 | r11i1p1f1 | Mauritsen et al. (2019) |
| MOHC | $\triangle$ UKESM1-0-LL | 250km | 85 | 85km | MetUM-HadGEM3-GA7.1 | r5i1p1f3 | Sellar et al. (2019) |
| MOHC | $\triangle$ HadGEM3-GC31-LL | 250km | 85 | 85km | MetUM-HadGEM3-GA7.1 | r5i1p1f3 | Roberts et al. (2019) |
| CNRM-CERFACS | $\bigcirc$ CNRM-CM6-1 ∗ | 250km | 91 | 78.4km | Arpege 6.3 | r2i1p1f2 | Voldoire et al. (2019) |
| CNRM-CERFACS | $\bigcirc$ CNRM-ESM2-1∗ | 250km | 91 | 78.4km | Arpege 6.3 | r1i1p1f2 | Séférian et al. (2019) |
| IPSL | $\bigcirc$ IPSL-CM6A-LR∗ | 250km | 79 | 40km | LMDZ | r1i1p1f1 | Boucher et al. (2020) |
| IPSL | $\bigcirc$ IPSL-CM5A2-INCA∗ | 500km | 39 | 80km | LMDZ | r1i1p1f1 | Sepulchre et al. (2020) |

# Appendix B: Variable Unit Transformation

To compare ERA5 and CMIP6 snowfall ($\mathrm{kg m^{-2} h^{-1}}$, Table 1) to CloudSat-CALIPSO ($\mathrm{sf_{CC}}$, we apply the following multiplication to achieve the satellite surface snowfall rate in $\mathrm{mm h^{-1}}$ (Table 1). The density of water ($\rho_{water}$) is about 1000 kg m$^{-3}$ and 1000 mm are one metre.

$$\mathrm{sf_{CC}} = \frac{mm}{h} \cdot \frac{1m}{1000mm} \cdot \rho_{water} = \frac{m}{h} \cdot \frac{1000kg}{1000m^3} = \frac{kg}{m^2 h} \tag{B1}$$

In the ERA5 reanalysis data, the daily mean of msr has units of $\mathrm{kg m^{-2} s^{-1}}$ (Table 1). To make ERA5 snowfall daily means comparable to CloudSat and CMIP6 snowfall rate, we apply the following multiplication to achieve the mean snowfall rate per hour (msr), where we know that one hour has 3600 seconds.

$$\mathrm{sf_{ERA}} = \frac{kg}{m^2 s} \cdot \frac{3600s}{h} = \frac{kg}{m^2 h} \cdot 3600 \tag{B2}$$

The same is done for the CMIP6 snowfall flux parameter (prsn, Table 1).

$$\mathrm{sf_{CMIP6}} = \frac{kg}{m^2 s} \cdot \frac{3600s}{h} = \frac{kg}{m^2 h} \cdot 3600 \tag{B3}$$

# Appendix C: Spatial distribution of $f_{sLCC}$ and $f_{snow}$

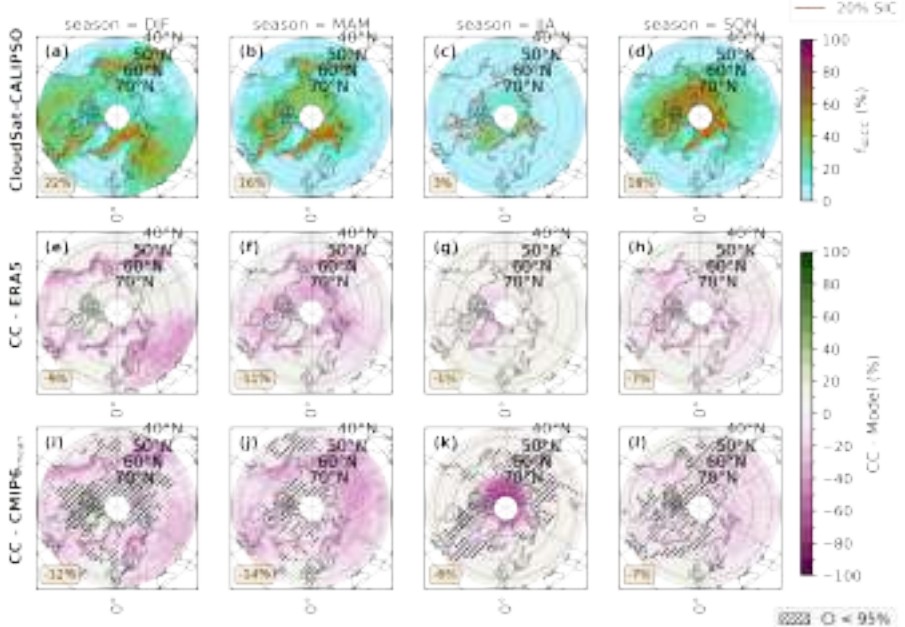

**Figure C1.** Seasonal averages of $f_{sLCC}$ in the NH mid-to-high latitudes between 2007 and 2010. Combined CloudSat and CALIPSO observations are shown in the first row (a-d). The last two rows are the difference plot. They are CloudSat-CALIPSO (CC) observations minus ERA5 (e-h) or CMIP6 model mean (i-l) where valid data occurs, with green (pink) values showing underestimation (overestimation) in ERA5 and the CMIP6 model mean concerning the satellite observations. Areas, where the difference between CloudSat-CALIPSO and CMIP6 model mean is not significant ($< 95\%$) are marked with hatches. The area-weighted averages for the study area where CloudSat-CALIPSO has observations are displayed in the lower-left corner of each map. The red line (in a-d) shows the average sea ice edge of $20\%$ sea ice concentration (SIC) between 2007 and 2010, for the given season.

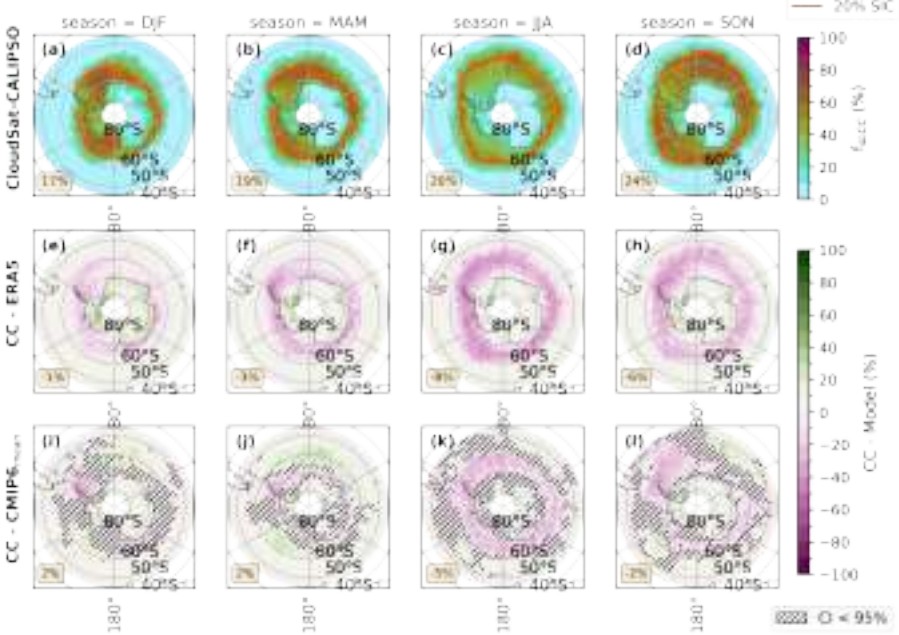

**Figure C2.** Seasonal averages of $f_{sLCC}$ in the SH mid-to-high latitudes. Layout and differences are identical to Figure C1.

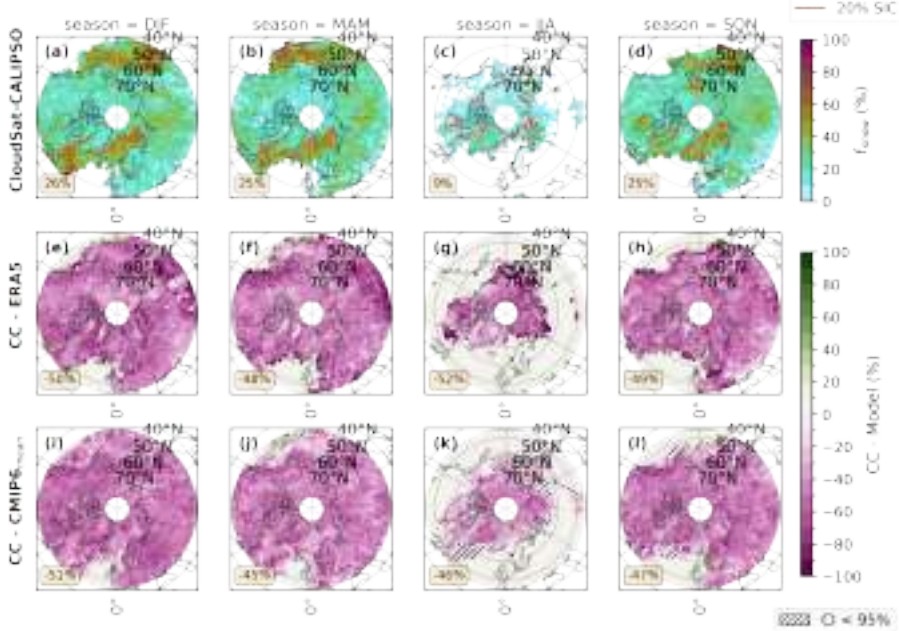

**Figure C3.** The figure presents the seasonal average of $f_{snow}$ in the NH. The layout and presentation resemble Figure C1.

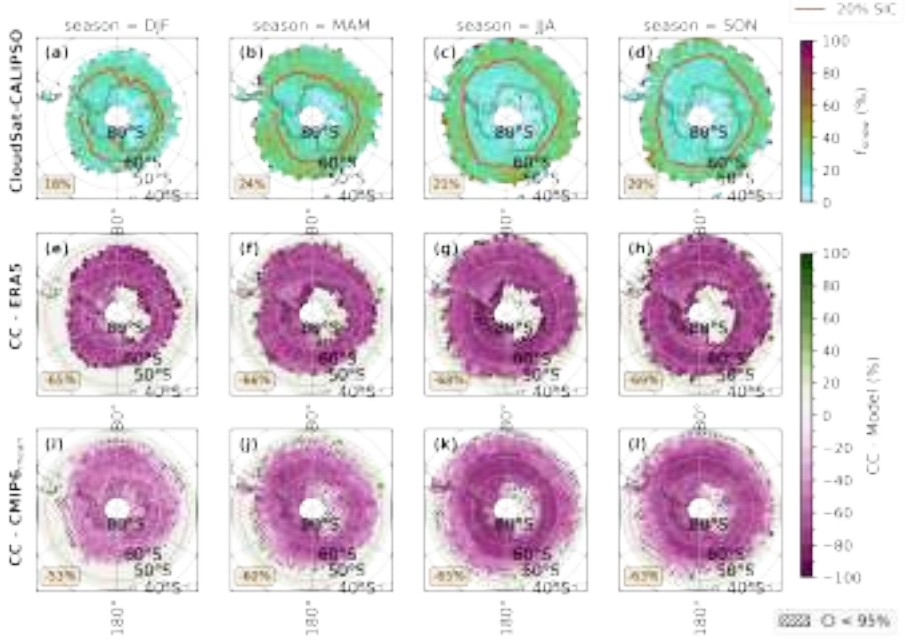

**Figure C4.** Seasonal averages of $f_{snow}$ in the SH. The layout and presentation resemble Figure C1.

**Appendix D:  Zonal mean of 2m temperature**

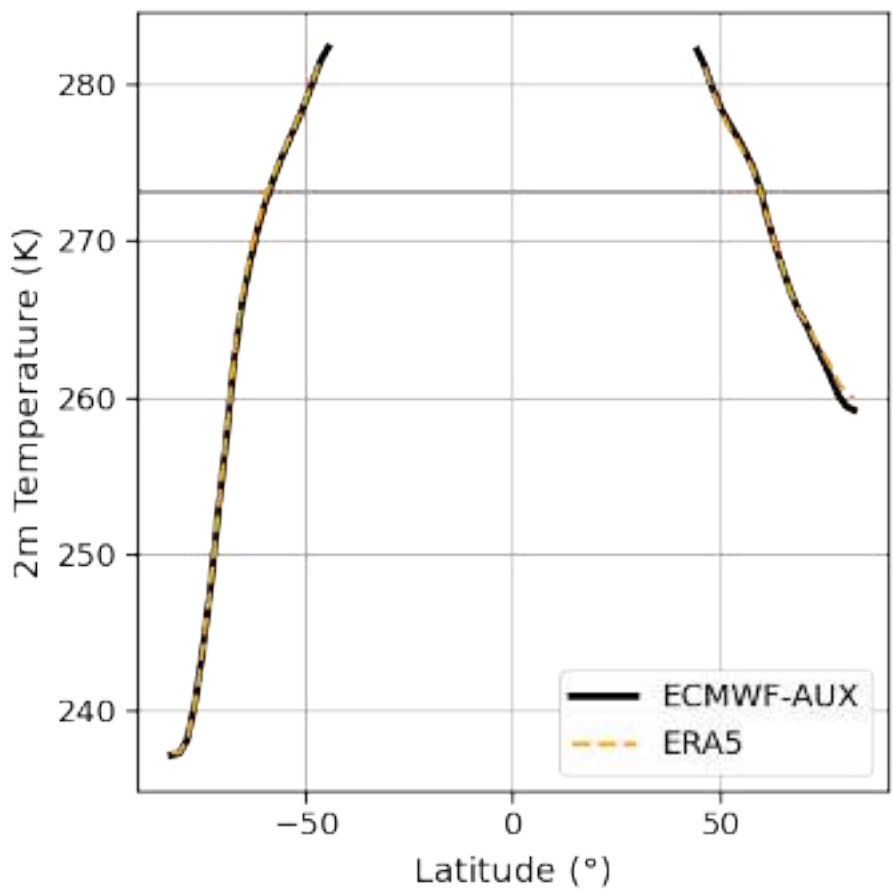

**Figure D1.** Zonal mean 2m temperature of ECMWF-AUX (black) and ERA5 (dashed-orange) for mid-to-high latitudes.

## Appendix E: LWP threshold sensitivity - SH

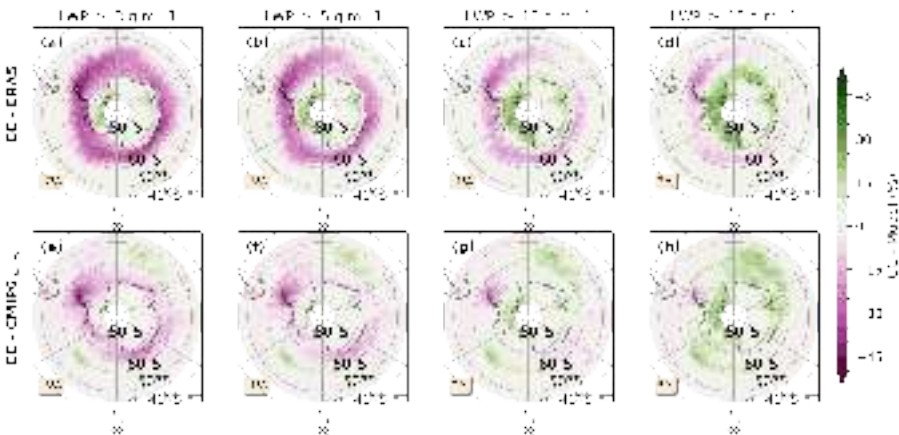

**Figure E1.** Annual average difference plots (CloudSat-CALIPSO (CC) minus model) for $f_{sLCC}$ in the SH mid-to-high latitudes. The first row represents ERA5 data (a-d), and the second row shows the CMIP6 model mean (e-h) difference. LWP thresholds applied of $3\mathrm{gm}^{-2}$ (a, e), $5\mathrm{gm}^{-2}$ (b, f), $10\mathrm{gm}^{-2}$ (c, g), and $15\mathrm{gm}^{-2}$ (d, h). The analysis in this study focuses on the LWP threshold of $5\mathrm{gm}^{-2}$. The maps in each row are accompanied by area-weighted averages for the study area, located in the lower-left corner of each map. The layout and area-weighted averages are calculated the same as those in Fig. 10.

**Appendix F: Supercooled liquid fraction in CMIP6 models**

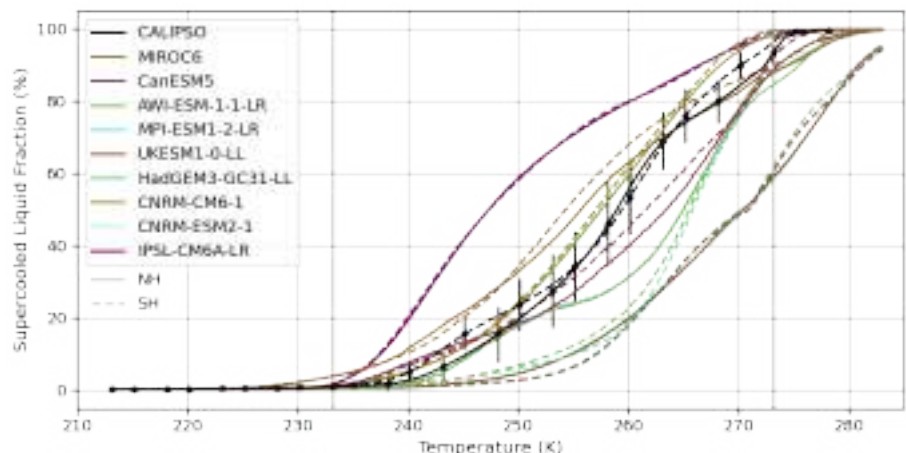

**Figure F1.** Supercooled liquid fraction (SLF) as a function of temperature for CALIPSO (black) and the CMIP6 models used in this study (color). Error bars on the CALIPSO SLF values correspond to one standard deviation. All values represent an area-weighted average for lat $\geq 45°$ in the Northern Hemisphere (NH, solid lines) and Southern Hemisphere (SH, dashed lines).

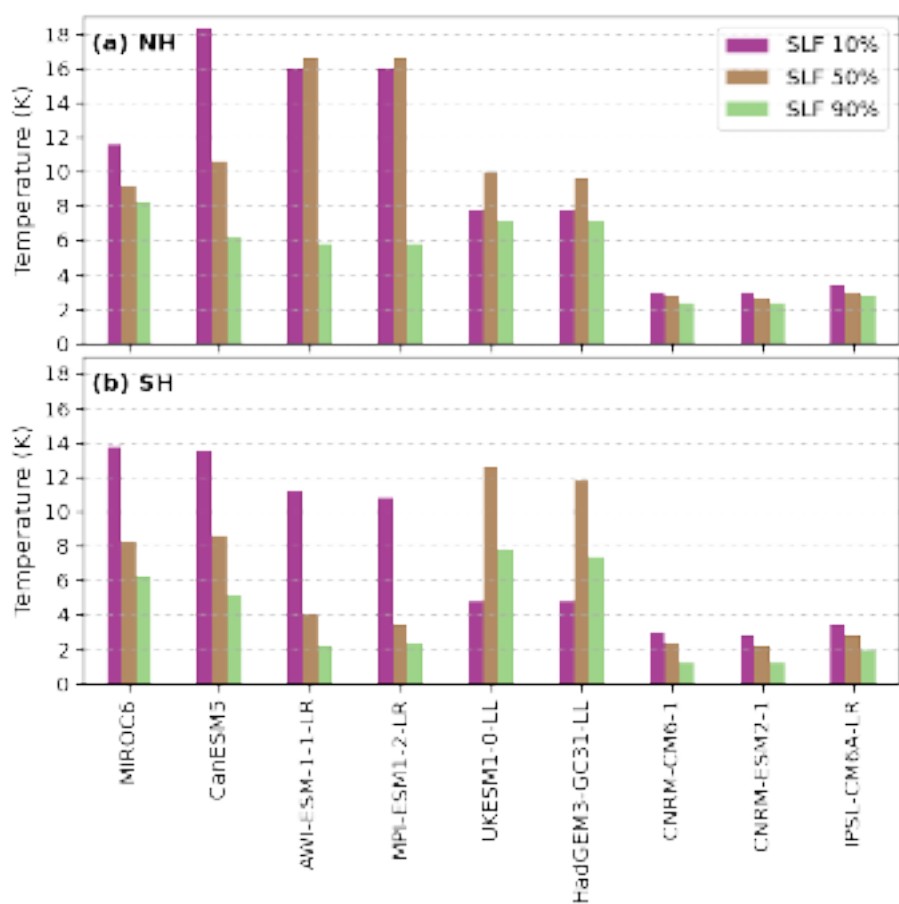

**Figure F2.** Interquartile range of the temperature at given supercooled liquid fractions (e.g., SLF 50%, representing 50% ice and 50% liquid) for the different models used in this study for lat $\geq 45°$ over the Northern Hemisphere (NH, a) and Southern Hemisphere (SH, b).

*Author contributions.* F.H., T.S., and A.S.D. designed the study. T.C. provided the satellite data and wrote parts of the Methods. H.C. performed the analysis to determine the supercooled liquid fraction as a function of temperature on average for each of the models used in this study. F.H. downloaded the ERA5 and CMIP6 data, analyzed the satellite, ERA5, and CMIP6 data, and wrote the manuscript. T.S. and A.S.D. supervised the project. T.S. provided the funding. All authors commented on several versions of the manuscript.

*Competing interests.* The authors declare that they have no conflict of interest.

*Acknowledgements.* This project has received funding from the European Research Council (ERC) under the European Union's Horizon 2020, and Horizon Europe research and innovation programs (Grant agreement numbers StG 758005, CoG 101045273, and GA 821205), and analyses were performed with the help of resources provided by Sigma2 - the National Infrastructure for High-Performance Computing and Data Storage in Norway. The color scheme for the scientific figures is used via the tool provided by Crameri (2018). In the writing process, Grammarly is used to review spelling and punctuation mistakes in the English text and ChatGPT is used to optimize python code where necessary. A.S.D. work is part of the ACCEPT project funded by the Norwegian Research Council (grant 315195). R.O.D. would also like to acknowledge EEARO-NO-2019-0423/IceSafari, contract no. 31/2020, and EU Horizon Europe research and innovation programme under grant agreement #101079385 for financial support. The authors thank Tristan L'Ecuyer, David Henderson, and Elin McIlhattan for their valuable discussions on processing the CloudSat-CALIPSO data.

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
