# Peer review of "Evaluation of Biases in mid-to-high latitudes Surface Snowfall and Cloud Phase in ERA5 and CMIP6 using Satellite Observations"

_EGUsphere, 2024_

## Author Response (AR1)

**Interactive comment on "Connection of Surface Snowfall Bias to Cloud Phase Bias - Satellite Observations, ERA5, and CMIP6" Hellmuth et al.**

**Reviewer #1**

This manuscript provides a well-organized and thorough analysis of snowfall biases associated with supercooled liquid-containing clouds (sLCCs). This topic is highly relevant for improving climate model prediction, particularly regarding climate feedbacks. The study takes a comprehensive approach, comparing satellite observations (CloudSat-CALIPSO), reanalysis data (ERA5), and global climate model simulations (CMIP6) to evaluate the physical characteristics of sLCCs and their snowfall production.

Key findings from the study reveal significant discrepancies: (1) ERA5 reanalysis and ten CMIP6 models consistently overestimate sLCC frequency and snowfall compared to CloudSat-CALIPSO observations; (2) Biases appear similar between ERA5 and CMIP6 models, suggesting cloud microphysics parameterization issues rather than meteorological discrepancies; (3) Accurately representing cloud phase and snowfall is crucial for reliable climate simulations and future climate projections. The biases, primarily found over polar regions, are not inconsequential and resolving them is essential for improving the fidelity of global climate models.

This work highlights the importance of improving cloud microphysics parameterizations in climate models for accurate representation of cloud phase and snowfall. While previous research has often focused on the underestimation of supercooled liquid in mixed-phase clouds, this paper demonstrates that sLCC frequency may be a more informative metric to explain model discrepancies in radiation and precipitation fields. This is a scientifically significant result that is important because observations reveal a separation of ice and liquid in time/space, whereas current models often simulate both phases coexisting in mixed-phase clouds.

Overall, the paper makes key contributions and provides valuable insights toward our understanding of clouds and their impacts on radiation and precipitation, well within the scope of this journal. The methodology and results are presented clearly and concisely. References are balanced and appropriately cited. Figures and tables are well-structured to effectively illustrate key research findings. Notably, all code used to produce figures using CloudSat, ERA5, CMIP6 analysis has been made publicly available as well. Based on the quality of research, strength of potential impact, and relevance to ACP, I recommend acceptance for publication.

*We thank the reviewer for their encouraging and positive comments on our work.*

**General Comments**

The authors of this manuscript leverage satellite observations of clouds and precipitation from CloudSat and CALIPSO instruments in order to evaluate 10 CMIP6 models and one reanalysis product (ERA5). The analysis region includes the mid to high latitudes and the variables of interest are (1) the frequency of supercooled-liquid containing clouds (fsLCC) and (2) how often the detected sLCCs produce snowfall.

Climate models have historically struggled to accurately simulate sLCCs, particularly in the high latitudes. The observational benchmarks provided by this manuscript are important for the ongoing process of improving both historical simulations and future predictions. The authors use these benchmarks to show that both ERA5 and the mean from the 10 CMIP6 models overestimate the frequency of sLCCs and the frequency with which sLCCs produce snowfall.

Overall, the manuscript is well organized, the figures and tables are well made, and the content will likely be of interest to a wide variety of scientists working on clouds/precipitation/climate modeling/etc.

My concerns mainly arise from some of the generalizations/conclusions presented. For example, the multi-model mean from the 10 CMIP6 models sometimes appears to be generalized to all CMIP6 models or even all ESMs. Also, the issue of model versions/generations is generally left out of the discussion of model biases. I recommend the authors revisit the discussion and conclusions to make clear the scope of their results and be more precise when putting them in the context of previous studies. I have included a list of specific suggestions below this summary that I hope will be of use to the authors in revising their manuscript.

We thank the reviewer for their positive and valuable comments which have greatly improved the manuscript. In the following, we address their concerns individually and provide the line numbers where the corresponding changes in the revised manuscript are made.

**Main Suggestions**

(1) Lines 291-293: "*We can safely assume the temperature to be similar between the ECMWF-AUX product used in CloudSat-CALIPSO and ERA5 daily mean, while atmospheric circulation and overall cloud cover should be well constrained by the observations used in the ERA5 reanalysis.*" I disagree that this is safe to assume. It is my understanding that the

ECMWF-AUX product is derived from a separate ECMWF dataset, AN-ECMWF, rather than ERA5. Given the discussion in the introduction regarding meaningful differences between ECMWF versions (lines 110-125), I believe additional analysis is necessary to support the authors' statement here, both regarding overall cloud cover and temperature. A figure comparing 2m temperature between the CloudSat-CALIPSO values and the ERA5 values as well as one that compares overall cloud cover would be necessary to rule out those differences as important factors in the sLCC and snowfall discrepancies.

The reviewer is correct that this statement was overconfident. We have now softened the language considerably to reflect that we cannot rule out differences also in temperature and cloud cover. However, we maintain that given the observations that are fed into the ECMWF-AUX data set (temperature, winds, humidity, etc), large errors in temperature and cloud macroscopic variables are much less likely than errors in microphysical variables that are largely unconstrained in the analysis. We have created the zonal mean 2m temperature of ECMWF-AUX and ERA5 and it can be seen, that the temperature of both datasets are quite close together, the latitudinal average difference is added to the text and is 0.24K±0.22K. We chose to compare the 2m temperature as the cloud cover is not included in the ECMWF-AUX data and added the figure to the appendix.

[Figure]

**Figure D1:** Zonal mean 2m temperature of ECMWF-AUX (black) and ERA5 (dashed-orange) for mid-to-high latitude.

**Lines 304 - 307:** "*It is reasonable to assume that the temperature in the ECMWF-AUX product used in CloudSat-CALIPSO is quite similar to the ERA5 daily mean. A comparison of the 2m temperature between ECMWF-AUX3 and ERA5 shows a latitudinal average difference of 0.24K ± 0.22K (Fig. D1). While atmospheric circulation and overall cloud cover should be well constrained by the observations used in the ERA5 reanalysis.*"

(2) Figure 5. It would be interesting/helpful to understand how this figure relates back to Figure 1. Is there a relationship between the location of sLCCs vs how likely they are to be snowing? For example, DJF observations (Fig.1a) show northern Europe with some of the most frequent sLCCs but producing snow the least often (Fig. 5a). Could this be an issue relating to the higher CloudSat bins used over land that the authors noted in the methods section (lines 174-176) or a characteristic sLCCs in that location?

Thanks for raising this point. Although it is possible that the higher minimum height bin of the radar used over land may lead to the 2C-SNOW-PROFILE product missing some of the precipitation over land, the fact that we see variability in the fsnow variable in regions where we expect similar cloud types (e.g. over North Central Russia) and to a lesser degree, that the models also show a lower fsnow over land, suggests that this land sea contrast is likely a representative feature. This is consistent with certain weather features such as cold air outbreaks, which once they reach a certain distance from the sea ice are frequently precipitating (e.g. Abel et al, 2017), while during the winter months, Northern Europe is frequently covered by stratus clouds, which are non-precipitating and supercooled (Cesana et al., 2012; McIlhattan et al., 2017).

We have now elaborated on this by stating:

"*While the influence of CAOs is also visible in the fsnow patterns across various seasons, with the exception of boreal summer, the fsnow values over land areas tend to be lower. Although it is possible that the higher minimum height bin of the radar used over land (~ 1000m) may lead to the 2C-SNOW-PROFILE product missing some of the precipitation over land, the variability in the fsnow values in regions where similar cloud types are expected suggests that this land-sea contrast is likely a representative feature. This discrepancy can be attributed to the limited moisture content and shallower boundary layer over land during winter, resulting in lower fsnow compared to ocean areas where CAOs provide deeper, moisture-rich clouds. Once clouds embedded in CAOs reach a certain distance from the sea ice, they frequently produce precipitation (e.g., Abel et al., 2017). In contrast, during the winter months, Northern Europe is often covered by non-precipitating, supercooled stratus clouds (Cesana et al., 2012; McIlhattan et al., 2017). The models also show a lower fsnow over land, supporting the idea that the observed land-sea contrast is a genuine characteristic rather than an artifact of measurement techniques.*" on **lines 352 - 362**.

(3) Lines 438 – 441: "*While previous studies have focused on the underestimation of supercooled liquid fraction (SLF) in mixed-phase clouds (Komurcu et al., 2014; Cesana et al., 2015; Tan and Storelvmo, 2016; Kay et al., 2016; Bruno et al., 2021; Shaw et al., 2022), this research focuses on the frequency of occurrence of sLCCs. This difference in the cloud phase metric can lead to seemingly contradicting conclusions.*" I encourage the authors to address the issue that the previous studies listed not only have different metrics, but also are

in some cases evaluating a different generation of climate models with markedly different cloud characteristics. Many models included in CMIP5 had too few sLCCs (e.g. Cesana, G. et al. (2012). Ubiquitous low-level liquid-containing Arctic clouds: New observations and climate model constraints from CALIPSO-GOCCP. Geophysical Research Letters, 39, L20804. https://doi.org/10.1029/2012GL053385). A problem that has been specifically addressed in some CMIP6 ESMs, resulting in a sizable increase in high-latitude cloud liquid (e.g. Lenaerts, J. T. M. et al. (2020). Impact of cloud physics on the Greenland Ice Sheet near-surface climate: a study with the Community Atmosphere Model. Journal Geophysical Research: Atmospheres, 125, e2019JD031470. https://doi.org/10.1029/2019JD031470). We thank the reviewer for pointing this out and fully agree. To address this omission we have now included some sentences to discuss the importance of different model versions on the potential differences between our observations and previous studies. This discussion has been added as follows on lines 475 - 488

"*Previous studies not only use different metrics but also, in some cases, evaluate different generations of climate models, each with distinct cloud characteristics. For instance, many models included in CMIP5 were found to have too few sLCCs (Cesana et al., 2012, 2015; Tan and Storelvmo, 2016; Kay et al., 2016; Lenaerts et al., 2017; McIlhattan et al., 2017). This deficiency was addressed in several of the next-generation models used in CMIP6, resulting in an increase in the simulated SLF (Mülmenstädt et al., 2021). Although some CMIP6 models used in this study still over- or underestimate the traditional SLF metrics compared to CALIPSO observations, models such as CNRM-CM6-1 and CNRM-ESM2-1 are within one standard deviation of the satellite observations (Fig. F1). Studies focusing on high latitudes have shown that comparing individual CMIP5 models with CMIP6 models reveal improved cloud representation, more closely aligning with observations due to microphysical adjustments in the newer model versions (Lenaerts et al., 2020; McIlhattan et al., 2020). Indeed, McIlhattan et al. (2020) showed that when going from the CMIP5 to the CMIP6 version of CESM the frequency of LCC bias switched direction and overestimates in comparison to CloudSat-CALIPSO. Lenaerts et al. (2020) showed that cloud coverage increased and is slightly overestimated in the CMIP6 version of CESM, but still underestimates ice water path. Consequently, some of the differences between our findings and those from previous studies may be attributed to these improvements in the newer model generations.*"

(4) Lines 441-442: "*We illustrate why our results do not necessarily contradict previous findings with the following example: …*" Following on my previous suggestion, I recommend trying to find a specific example from the CMIP6 models included in this study and seeing if the same generation of that model is also evaluated in one of the listed previous studies. The statements in this section of the discussion seem to be attributing discrepancies only to differences in the metric (sLCC frequency vs SLF) and not addressing the important issue of model version.
In light of this and the previous comment, we have now added specific examples of how the changes incorporated by various modelling centres lead to differences in the representation of cloud properties across CMIP generations. See the previous response to see the added text on lines 475-488.

(5) Lines 451-453: "*In contrast, McIlhattan et al. (2017) showed that the CESM-LE underestimates the fLCC by ~ 17% and overestimates the fsnow by ~ 57% in the Arctic. However, since we utilize a different metric (sLCC instead of LCC), there is no reason to expect the model biases to be identical.*" From my understanding of the two metrics, fLCC and fsLCC wouldn't be likely to produce biases with the opposite sign in the high-latitudes (outside of perhaps summer) if the models contained similar cloud systems. It seems more likely that the difference between this study's biases and those found by and McIlhattan et al. (2017) arise from differences in the models' cloud systems. The same metrics from McIlhattan et al. (2017) were used to evaluate LCCs in CESM2 (a CMIP6 generation model; McIlhattan, E. A., et al. (2020). Arctic clouds and precipitation in the Community Earth System Model version 2. Journal of Geophysical Research: Atmospheres, 125, e2020JD032521. https://doi.org/10.1029/2020JD032521), and those findings appear to be more similar to those presented here. I strongly encourage the authors include the issue of model generation when comparing their results to earlier studies.

The following figures show the fLCC (a-d) as defined by McIllhattan et al. (2017) and the fsLCC (e-h) as defined in our study. The lower panel in the figures (i-l) show the difference between the two metrics. fLCC and fsLCC are especially different over the ocean.

[Figure]

[Figure]

We included the additional reference given by the reviewer and added some sentences to the discussion on model versions. Specifically, we added the following sentences in **Lines 492-509**:

"*However, since we utilize a different metric (sLCC instead of LCC), there is no reason to expect the model biases to be identical. Nevertheless, in a further study by McIlhattan et al. (2020) it was shown that the LCC frequency in the newer CESM version is more aligned with the satellite observations except for in the summer months, where it overestimates the LCC frequency.*

*Similarly, we generally observe that ERA5 and the CMIP6 mean overestimate the fsLCC to various extents during all seasons. These overestimations are likely linked to the microphysical parametrizations of cloud processes that govern cloud phase. This finding aligns with McIlhattan et al. (2020), indicating that while newer model versions have advanced in representing LCC frequencies more accurately, they can still overestimate cloud occurrences, particularly in specific seasons. Precipitation in the new CESM version is more frequent but lighter overall compared to the previous version, which is similar to our findings indicating that the models's sLCCs produce continuous snowfall, analogous to the "perpetual drizzle" problem (Mülmenstädt et al., 2020; Lavers et al., 2021).*

*Furthermore, while McIlhattan et al. (2017, 2020) considered a single ESM, our study considers an ensemble of CMIP6 models. Nevertheless, the insights from McIlhattan et al. (2017, 2020) provide relevant context for our finding that ERA5 and the CMIP6 model mean produce sLCCs more frequently than observed in the NH and SH mid-to-high latitudes,*

*especially over the sea ice and land depending on the season (Figs. C1 and C2). In these regions, not only is the frequency of occurrence of sLCCs too high, but the sLCCs are too efficient at producing snowfall in ERA5 and CMIP6 models (Figs. 5 and 6). The latter finding is consistent with the findings of McIlhattan et al. (2017) that LCCs produce snow too frequently in the CESM-LE model.*"

(6) Lines 471-475: "*However, the fact that very similar biases are found in the reanalysis as in the CMIP6 models indicates that this is not the explanation for the identified biases, as weather patterns and surface conditions in ERA5 should be very close to the observed. This is an important conclusion that is possible because both ERA5 and CMIP6 models were included in the present analysis. However, we cannot rule out a modest contribution to the biases from circulation differences, even for the reanalysis.*" Reanalysis products are known to be less reliable at high latitudes due in part to sparse observations (e.g. Liu, Y., and J. R. Key, 2016: Assessment of Arctic Cloud Cover Anomalies in Atmospheric Reanalysis Products Using Satellite Data. J. Climate, 29, 6065–6083, https://doi.org/10.1175/JCLI-D-15-0861.1). Also, it is clear from the authors' Figure 4 that the individual models within this CMIP6 subset do not all have biases matching ERA5. So, I do not see a clear reason to conclude that differences in weather patterns and/or surface conditions are not important contributors to the model or reanalysis biases. I recommend revising or removing this language.

We revised the text to include the reference given by the reviewer and to include the individual model variation. **Lines 521 - 531** now reads:

"*However, the fact that very similar biases are found in the reanalysis as in the CMIP6 models might suggest that surface conditions and weather patterns are not the main explanation for the identified biases. That being said, reanalysis products are known to be less reliable at high latitudes (Liu and Key, 2016). We therefore cannot rule out differences in weather patterns as a partial cause of the biases discussed above, but maintain that deficiencies in the model representation of cloud microphysics is the more likely culprit. The inclusion of both ERA5 and CMIP6 models in this analysis is thus beneficial as it helps highlight potential root causes of the identified biases, although it should be noted that individual CMIP6 models do not all show biases that match ERA5 (Figs. 4 and 8). As stated above, we cannot entirely rule out a modest contribution to the biases from circulation differences, even for the reanalysis. For example, according to Boisvert et al. (2020), all reanalysis products show lower precipitation amounts in the southern Ross Sea and Weddell Sea embayments near the ice shelves. They explain that this is due to persistent cold and dry katabatic wind blowing from the continent across the ice shelves and out over the sea ice, which is not beneficial for precipitation formation and not captured well by the reanalyses.*"

(7) Section 5.2 "*Implications for modeling and future projections.*" The second and third paragraphs of this section include primarily very broad generalizations that do not tie directly to the results of the study. Consider revising to focus more clearly on the impact and implications of this paper's results.

We agree that our initial implications were quite broad and general. To more directly link the implications to our study we have revised the two paragraphs starting on **Lines 572 - 596** as follows:

"*The overestimations of fsLCC and fsnow could stem from how models handle cloud microphysical processes, particularly the WBF process and the partitioning of cloud condensates into the liquid and ice phase. Some identified biases in ESMs may be due to the specific parametrization schemes used in the models, which could rely on simple temperature-dependent cloud phase partitioning. For example, models with a temperature-dependent cloud phase could simulate a fixed liquid fraction at a given temperature within the mixed-phase range, making it impossible for them to simulate an all-ice or all-liquid cloud in that range. Consequently, these models could tend to under- or overestimate the occurrence of sLCCs.*

*Interestingly, models with more sophisticated microphysics schemes (Figs. F1 and F2) do not necessarily perform better (Figs. 4 and 8). Larger biases in cloud phase can still occur, for example if these models do not accurately represent INPs. This underscores how generational advancements and implementations of observational constrains can significantly influence the comparison and interpretation of cloud and precipitation patterns.*

*Continuing work should be performed on the comparison between ESMs, reanalysis and observations at specific locations to better understand microphysical processes and their representation in ESMs and reanalysis data. Cloud microphysical scheme development should, therefore, focus on improving these processes in order to reduce cloud phase and surface snowfall biases. More sophisticated schemes should be explored, and ensemble studies with varied microphysical model parameterizations can provide insights into how different parameterizations affect the representation of sLCCs and snowfall in climate simulations.*

*Ground and aircraft observations in connection with field campaigns at specific locations should, therefore, be increased, especially to improve our understanding of cloud phase, liquid and ice water content, and cloud top temperatures in regions where sLCCs form. These field campaigns should also focus on the processes responsible for snowfall from these clouds.*

*Finally, as shown in the present study, satellite observations are an essential validation tool for assessing the representations of clouds and precipitation in ESMs. To ensure that future model generations are as accurate as possible, continuous improvements to satellite retrievals are required to further refine our understanding of cloud and snowfall properties, especially in high-latitude regions where limitations in sun angle and ground-track are prevalent. The accurate representation of these cloud and snowfall processes in models, validated through enhanced satellite observations, is essential for reliable simulations of weather and climate. This accuracy is particularly important for making future predictions of the hydrological cycle, which are essential for understanding and mitigating the impacts of climate change.*"

**Minor Suggestions**

(8) Title: It might be helpful to include the region in the title (e.g. mid-to-high latitudes) to help the paper reach the appropriate audience.

Thank you for pointing this out. We have now added the study region as you have suggested in the title so that it reads:

"*Evaluation of Biases in mid-to-high latitudes Surface Snowfall and Cloud Phase in ERA5 and CMIP6 using Satellite Observations*"

(9) Line 16-21: I suggest re-writing the opening paragraph of the introduction to improve flow and to be simpler and more declarative.

*We agree and have now rewritten the introduction paragraph and also included some more specifics about what is meant by ecosystems as recommended below. The paragraph now reads:*

*"Snowfall and snow cover have a significant impact on the Earth's energy budget and the hydrological cycle, especially in mid- and high-latitudes, and thereby strongly impact ecosystems and human societies. Snow cover influences vegetation growth, animal populations, and ecosystem processes, while also impacting economic activities, infrastructure, and health (Callaghan et al., 2011; Bokhorst et al., 2016). Indeed, snowfall is also an important water resource globally (Barnett et al., 2005) and snow cover increases the surface albedo, reducing the absorption of incoming solar energy (Zhang, 2005). However, heavy snowfall events have the potential to negatively impact local communities and infrastructure (Eisenberg and Warner, 2005; Scott et al., 2008; Fox et al., 2023)." on* **Lines 16-22**.

(10) Line 17-18: "*…strongly influence ecosystems and human societies in these regions…*" I suggest the authors be more specific here and include citations.

In line with the previous comment, we have now rewritten the Introductory paragraph and included more details about what is meant by ecosystems and human societies. Please see the previous comment for the updated text.

(11) Line 22: "*Snowfall is directly linked to the cloud phase,*" What aspect of snowfall is directly linked to the cloud phase? Rate? Frequency? It seems from the sentences that follow that snowfall at high latitudes is produced by both sLCCs and pure ice clouds, however, the specific link between the cloud phase and the snowfall is unclear. Consider revising.

Thanks for pointing this out and we agree. We have now changed the first sentence of this paragraph to "*Cloud phase determines the formation process of snow*" on **line 23**.

Additionally, we have now added a sentence that specifically states how cloud phase relates to snowfall on **lines 26-28** that reads:

"*The cloud phase affects not only the rate and intensity of snowfall but also the microphysical properties of the snowflakes that reach the ground (Jiusto and Weickmann, 1973; Liu, 2008).*"

(12) Lines 38-41: "*Several studies, including Murray et al. (2021) showed that increased temperatures, especially in polar regions, have caused a shift of mixed-phase clouds towards higher latitudes and altitudes due to the ice reduction in the atmosphere in these regions. The shift in cloud phase towards more liquid and less ice leads to a reduction in the fraction of precipitation falling as snow, resulting in an expected decrease in snowfall events and duration of the snowfall season for most regions in the Northern Hemisphere (NH, Danco et al., 2016; Chen et al., 2020).*" It is my understanding that mixed-phase clouds in the high latitudes (particularly the sLCCs that were introduced earlier) primarily produce snowfall, so I would not necessarily expect a snowfall decrease with an increase in mixed-phase clouds. Please clarify and include specific supporting results from the papers cited here.

Good catch, we have now clarified what we meant here by adding more details from Chen et al., (2020) and Danco et al., (2016) on **Lines 42-46**.

"*The shift in cloud phase towards more liquid and less ice leads to a reduction in the fraction of precipitation falling as snow in previously snowy areas in the Northern Hemisphere (NH), but snowfall events will happen further north in the future (Chen et al., 2020). The shift in isolation would result in an expected decrease in snowfall events and duration of the snowfall season for most regions in the NH (Danco et al., 2016; Chen et al., 2020).*"

(13) Lines 84-85: "*Previous studies have contributed to a better understanding of the uncertainties associated with satellite measurements of clouds and precipitation*" I suggest the authors be more specific on what those uncertainties are. What is the general understanding of the limitations of satellite measurements of clouds and precipitation and the magnitudes/signs of the expected biases in those measurements?

We have now added some sentences about the specific uncertainties highlighted in the Stephens and Kummerow (2007) and Hiley et al., 2011 studies on **Lines 90-95**.

"*For example, Stephens and Kummerow (2007) identified two primary sources of uncertainty in retrieval methods, errors in distinguishing between cloudy and clear sky scenes and between precipitating and non-precipitating clouds. Furthermore, the forward models used are highly sensitive to their input parameters, particularly the radiative transfer and atmospheric models. Hiley et al. (2011) demonstrated that snowfall retrievals are also influenced by retrieval assumptions and the use of different ice particle models, which can significantly affect the estimated snowfall rates.*"

(14) Lines 195-197: "*We incorporate tcrw with the 2t threshold below 0∘C, as this threshold is used to exclude any rainwater below the melting layer. By using tclw and tcrw and applying the temperature threshold, we can analyze the role of supercooled liquid water within clouds and the contribution of liquid water to the snowfall precipitation process in ERA5.*" Are the authors excluding all hourly LWP values with instantaneous hourly 2 m temperature greater than 0C or excluding all daily mean LWP values where the daily mean 2 m temperature is

greater than 0C? It is unclear to me if the threshold is applied before or after making the daily average.

We agree that this was unclear as written. To clarify, the 2 m temperature threshold is applied to the daily mean values rather than the instantaneous hourly values. We have now adjusted the sentence on **Lines 207-210** to:

"*We incorporate tcrw with the T2m threshold below 0°C, as this threshold is used to exclude any rainwater below the melting layer. By using daily mean values of tclw and tcrw and applying the daily man temperature threshold, we can analyze the role of supercooled liquid water within clouds and the contribution of liquid water to the snowfall precipitation process in ERA5.*" to make this clearer.

(15) Lines 223-225: "*The slight mismatch in the time range is of limited relevance, as CMIP6 model simulations are not designed to reproduce the exact temporal evolution of past weather*" It is my understanding that the CMIP6 models incorporate historical forcings so are designed to produce fairly representative simulations for the historical record. I agree that it is likely the mismatch will not change the authors' conclusions, but perhaps it would be worthwhile to compare results from 2006-2009 to results from 2007-2010 for a model that has all years available to determine the magnitude of the difference.

The reviewer is correct that the CMIP6 historical simulations do incorporate prescribed natural and anthropogenic historical forcings. Unlike weather forecasts, these simulations are not periodically updated with new climate state information. Instead, CMIP6 historical simulations follow the prescribed forcings and are intended to reproduce observed multi-decadal climate statistics rather than exact sequences of weather and climate events. However, the forcing difference between 2006-2009 and 2007-2010 is negligible and would not lead to noticeable differences between the two simulated periods. The difference would instead originate from differences in internal variability between the two periods, which would in neither case resemble the actual internal variability. Therefore, we argue that the comparison between 2006-2009 and 2007-2010 for selected CMIP6 models would not be meaningful in this context.

(16) Figure 4: "*The heatmap colors correspond to the absolute differences of area-weighted averages…*" I'm curious the reason behind using absolute difference instead of something like mean difference that would highlight the direction of the bias rather than just the magnitude. I think the figure may be more useful to readers if the direction of the bias was more clear.

That is a great suggestion. We have now updated Fig. 4 and Fig. 8 to show the direction and magnitude of the model bias. We also adjusted the text by removing "absolute" in the corresponding sections.

[Figure]

**Figure 4**. Magnitude of seasonal area-weighted averages (between |45° − 82°|) of $f_{sLCC}$ for CloudSat-CALIPSO, ERA5, CMIP6 models (numbers). The heatmap colors correspond to the differences of area-weighted averages of $f_{sLCC}$ between CloudSat-CALIPSO and ERA5 and CMIP6 models. Green (pink) values indicate underestimation (overestimation) of the individual model with respect to CloudSat-CALIPSO. (a) for the mid-to-high latitude NH, and (b) for the mid-to-high latitude SH. Per season, the smallest (largest) seasonal and spatial differences are outlined with blue (red) lines.

[Figure]

**Figure 8**. Magnitude of seasonal area-weighted averages (between |45° − 82°|) of $f_{snow}$ for CloudSat-CALIPSO, ERA5, CMIP6 models (numbers). The heatmap colors correspond to the differences of area-weighted averages of $f_{snow}$ between CloudSat-CALIPSO and ERA5 and CMIP6 models. Green (pink) values indicate underestimation (overestimation) of the individual model with respect to CloudSat-CALIPSO. (a) for the mid-to-high latitude NH, and (b) for the mid-to-high latitude SH. Per season, the smallest (largest) seasonal and spatial differences are outlined with blue (red) lines.

(17) Lines 339-340: "*While the signature of CAOs is also visible in the fsnow patterns, besides in boreal summer, land areas have lower values as these areas warm in response to increased insolation.*" I am having difficulty understanding this statement, consider revising for clarity and perhaps include references to specific figure panels.

Thanks for pointing this out and we agree that it was not clear what we meant here. We have now rephrased the sentence on **Lines 352-362** to read:

"*While the influence of CAOs is also visible in the fsnow patterns across various seasons, with the exception of boreal summer, the fsnow values over land areas tend to be lower. Although it is possible that the higher minimum height bin of the radar used over land (~ 1000m) may lead to the 2C-SNOW-PROFILE product missing some of the precipitation over land, the variability in the fsnow values in regions where similar cloud types are expected suggests that this land-sea contrast is likely a representative feature. This discrepancy can be attributed to the limited moisture content and shallower boundary layer over land during winter, resulting in lower fsnow compared to ocean areas where CAOs provide deeper, moisture-rich clouds. Once clouds embedded in CAOs reach a certain distance from the sea*

*ice, they frequently produce precipitation (e.g., Abel et al., 2017). In contrast, during the winter months, Northern Europe is often covered by non-precipitating, supercooled stratus clouds (Cesana et al., 2012; McIlhattan et al., 2017). The models also show a lower fsnow over land, supporting the idea that the observed land-sea contrast is a genuine characteristic rather than an artifact of measurement techniques.*"

(18) Lines 375-376: "*ESMs have too many sLCCs*" I suggest changing the language here, since the results show only that the CMIP6 multimodel mean has too many sLCCs. There are certainly individual ESMs that produce too few, indicated by the green dots in this manuscript's Figure 3.
That is a fair point. We have now revised the first paragraph in section 4 "Sensitivity tests" (**Lines 396-405**), accordingly:

"*In the previous section, we examined the frequency of occurrence of sLCCs (fsLCC) and the frequency of occurrence of surface snowfall (fsnow) from sLCCs in CloudSat-CALIPSO, ERA5, and CMIP6 data. We found that the reanalysis and ESMs overestimate sLCCs and the frequency of surface snowfall in comparison to CloudSat-CALIPSO. These biases in the CMIP6 mean have potentially significant implications for our ability to predict how sLCCs and snowfall might change with future warming. At the same time, it is important to note that although the CMIP6 multimodel mean has this overestimation, some of the individual ESMs are much closer to the observations (green dots in Fig. 3), suggesting that these members may have more representative changes in clouds and snowfall in the future. It is therefore important to ensure that these findings are robust, and not overly reliant on subjective decisions or limitations in the design of the comparison. In the following subsections, we investigate whether the identified discrepancies between the observations on one hand and ERA5 and the CMIP6 models on the other can be explained by sampling biases or instrument sensitivity.*"

The same concern/suggestion goes for line 425: "*This would most likely also hold for the CMIP6 models, although this cannot be confirmed.*"
We extended our statement to a softer language. **Lines 448-451** read:

"*However, the conclusion that ERA5 overestimates the fsLCC does not change by changing to a higher time resolution (Fig. 9). While the sensitivity analysis for ERA5 data indicates that changes in output frequency do not significantly affect the results, the same conclusions cannot be directly extrapolated to CMIP6 models. However, based on our analysis, it is reasonable to assume that similar results could be observed for individual ESMs within CMIP6, although this cannot be confirmed.*"

(19) Lines 385-388: "*Milani et al. (2018) found that applying adjustments and a temperature threshold to the CloudSat snowfall retrieval led to a decrease in the estimated occurrence of snowfall events, primarily in the ocean regions surrounding Antarctica. Although these adjustments did not have the same effects everywhere, this highlights the sensitivity of the CloudSat retrievals to the assumptions made within them.*" Consider including the magnitude

of the decrease in events Milani et al. (2018) found in order to give readers an idea of the magnitude of uncertainty others have found coming from the observational data.
Thanks for this suggestion, we have now added the magnitude to the text on **Lines 409-411**:

"*Milani et al. (2018) found that applying adjustments and a temperature threshold to the CloudSat snowfall retrieval led to a decrease in the estimated occurrence of snowfall events by up to 30%, primarily in the ocean regions surrounding Antarctica.*"

(20) Lines 486-488: "A*s discussed above, many models have a simple temperature-dependent cloud phase that would almost certainly cause them to overestimate the fsLCC and $f_{snow}$.*" It would be useful to indicate which of the 10 models included in the study have this simple temperature dependency. Also, it seems that the specific temperature threshold used in a given model would influence the bias as well, are all the cloud phase temperature thresholds the same?
We completely agree and were hoping to look into this more. However, it is not immediately clear from the model documentation which models use a purely temperature-dependent partitioning between liquid and ice. As a quick check though, we have investigated this by plotting the supercooled liquid fraction as a function of temperature on average for each of the models used in this study **(see Figure F1 below)**. If the models relied on something other than temperature i.e. aerosol, then there would be a difference between the NH (solid lines) and SH (dashed lines) in the supercooled liquid fraction. As most of the models have these two lines on top of each other, it is likely that they are only using temperature-dependent partitioning.  Therefore, we assume that MIROC6, CanESM5, AWI-ESM-1-1-LR, MPI-ESM1-2-LR, UKESM1-0-LL, and HadGEM3-GC31-LL are using a more complex phase partitioning while other models like CNRM-CM6-1, CNRM-ESM2-1, and IPSL-CM6A-LR are assumed to have a simple temperature-dependent cloud phase.

[Figure]

**Figure F1:** Supercooled liquid fraction (SLF) as a function of temperature for CALIPSO (black) and the CMIP6 models used in this study (color). Error bars on the CALIPSO SLF values correspond to one standard deviation. All values represent an area-weighted average for lat ≥ 45° in the Northern Hemisphere (NH, solid lines) and Southern Hemisphere (SH, dashed lines).

This assumption is further supported by the interquartile range (IQR) of the temperature at various supercooled liquid fractions as shown in **Figure F2**, where both CNRM model versions and the IPSL model, have a much smaller IQR at the given SLFs than the other models. Despite the fact that MIROC6, CanESM5, AWI-ESM-1-1-LR, MPI-ESM1-2-LR, UKESM1-0-LL, and HadGEM3-GC31-LL have a more variable SLF-temperature relationship, from Fig. 4 it is not clear if these models perform better with respect to fsLCC. This analysis is part of a follow-up study by co-author Haochi Che.

[Figure]

**Figure F2:** Interquartile range of the temperature at given supercooled liquid fractions (e.g., SLF 50%, representing 50% ice and 50% liquid) for the different models used in this study for lat ≥ 45∘ over the Northern Hemisphere (NH, a) and Southern Hemisphere (SH, b).

Since there is no clear improvement in fsLCC (Fig. 4) whether the model has a purely temperature-dependent partition or not, we have decided to make adjustments to the text. **In lines 541-548:**

*"A follow-up study will investigate the SLF by isotherm for mixed-phase clouds of several CMIP6 models. Figure F1 shows that some of the CMIP6 models have a more variable SLF in time and space (e.g., AWI-ESM-1-1-LR, MPI-ESM1-2-LR), which might be an indication that they could have more sophisticated microphysics and could, therefore, perform better in terms of sLCC and surface snowfall occurrence. However, it is not immediately clear which models use a purely temperature-dependent cloud phase partitioning. From the interquartile range (IQR, Fig. F2) CNRM-CM6-1, CNRM-ESM2-1, and IPSL-CM6A-LR are assumed to have a simple temperature-dependent cloud phase due to the smaller IQR of the temperature at various SLF. Interesting to note, these models with the simpler temperature-dependent schemes are the most poorly performing models for fsnow (Fig. 8)."*

**Typos/Grammatical Issues/Word Choice/Needs Citation:**

(21) Line 27: *"among others,"* It is unclear what "others" the authors mean here, consider revising.
We substituted the phrase "*among others*" with "*partly*" in **Line 29**.

(22) Line 69: "*ESMs have previously been shown to not accurately represent cloud phase*" consider adding a regional descriptor to this statement (e.g. global, Arctic, high-latitude), I believe the majority of the citations for this sentence deal specifically with high latitude clouds.
There are several studies that have also focused on the global underestimation of LWP and overestimation of IWP. But the studies we primarily focus on are related to the high-latitude regions. Therefore we rephrased it to:

*"ESMs have previously been shown to not accurately represent cloud phase by often underestimating liquid and overestimating ice, compared to satellite measurements. This has particularly been the case for high-latitude regions (Komurcu et al., 2014; Cesana et al., 2015; Tan and Storelvmo, 2016; Kay et al., 2016; McIlhattan et al., 2017; Bruno et al., 2021; Shaw et al., 2022)."* on **Lines 72-74**.

(23) Line 76: "*must lead to biases*" consider softer language, since other model biases could compensate and lead to correct precipitation simulation (two wrongs making a right).
We have replaced "*must*" with "*could*" to soften the language (**Line 80**) and added an additional sentence in **lines 81-82**, following for **lines 78-82**:

*"Because cloud phase and snowfall are tightly linked through the processes outlined earlier, any inaccuracies in representing cloud phase could lead to biases in the simulation of snow growth, formation, and the precipitation reaching the ground in solid or liquid form (Mülmenstädt et al., 2021; Stanford et al., 2023). It is important to note, however, that while such biases in cloud phase representation might exist, other compensating model biases could nevertheless lead to accurate precipitation simulations."*

(24) Line 84: "*allowing for continuous monitoring*" since the overpasses are every 16 days, this seems periodic rather than continuous, please clarify.
Thanks for catching this. It has now been clarified on **Lines 87-88**:

"*The CloudSat-CALIPSO constellation overpasses the same regions approximately every 16 days, providing long-term periodic monitoring of cloud and snowfall characteristics across the globe.*"

(25) Lines 88-89: "*Previous studies have shown that ESMs produce double the amount of snowfall relative to satellite observations (Heymsfield et al., 2020).*" Consider softer language. As written, this implies multiple studies have shown that all ESMs produce 2x measured snowfall.
We rephrased the sentence using softer language to make the statement more general. **Lines 97-99** now read:

"*Previous studies suggest that there is a notable difference in snowfall estimates between ESMs and satellite observations. For example, (Heymsfield et al., 2020) found that the Met Office Unified Model and the Community Atmosphere Model 6 produce double the amount of snowfall relative to satellite observations.*"

(26) Lines 116-118: "*In the Arctic, the ERA-Interim data qualitatively represented the interannual snowfall rates and seasonal cycle well but underestimated high snowfall rates significantly during summer and overestimated weak snowfall rates over open water compared to CloudSat.*" Please cite.
Thank you for catching this. The citation for Edel et al., 2020 is now added to the statement. (**Lines 127-129**)

"*In the Arctic, the ERA-Interim data qualitatively represented the interannual snowfall rates and seasonal cycle well but underestimated high snowfall rates significantly during summer and overestimated weak snowfall rates over open water compared to CloudSat (Edel et al., 2020).*"

(27) Line 150: "*Our primary emphasis is on snow-producing sLCCs*" From the end of the introduction, it seems the authors are interested in snow-frequency from sLCCs, so wouldn't the emphasis be on all sLCCs and determining how often/where they are producing snow? Please clarify.
Thank you for pointing this out. As suggested, we have now rephrased the sentence on **Lines 162-163** to:

"*Our primary emphasis is on sLCCs in the mid-to-high latitudes and understanding how frequently and where they produce snowfall.*"

(28) Line 192 vs Line 203: "*sLCC*" vs "*LCC*" there are some consistency issues between using "super-cooled liquid containing clouds" and "liquid containing clouds" I would guess since the temperature threshold has been applied, it should be sLCC on line 203.

We agree that this nomenclature is confusing so have changed "LCC" to "sLCC" in this case (**Lines 216-217**), thanks.

"*Nevertheless, a non-zero model daily mean LWP means there was an sLCC in the grid box at some point during the day.*"

(29) Figure 9: Consider including column headings. While the information is included in the caption, it would be helpful to have it on the columns as well.

Thanks, the column headings have been added in Figure 9 as shown below:

[Figure]

*Figure 9*. *NH mid-to-high latitude monthly averages of fsLCC for July (a-e) and November (f-j) 2007 to 2010. The columns show the CloudSat-CALIPSO (CC) monthly means (a, f), ERA5 monthly means based on daily average values (b, g), and ERA5 monthly means based on hourly values (c, h). The fourth and fifth columns are the difference plots between CloudSat-CALIPSO and the ERA5 daily averages (d, i) and ERA5 hourly values (e, j), respectively. In the last two columns, green (pink) values indicate underestimation (overestimation) with respect to the reference used. Area-weighted averages for the study area (45∘N − 82∘N) are located in the lower-left corner of each map and exclude the dotted area (in b, c, g, h).*

(30) Lines 423-424: "*reduces from 12% overestimation to 7% in November depending on the region.*" Since the 12 and 7% values are means for the whole region, I suggest removing "*depending on the region*".

Good catch, we have now removed "depending on the region" on **Lines 446-447**.

"*The area-weighted difference between CloudSat-CALIPSO and ERA5 hourly values is negligible in July (1%) and ~ 15% in the inner Arctic and reduces from 12% overestimation to 7% in November.*"

---

## Author Response (AR2)

**Interactive comment on "Evaluation of Biases in mid-to-high latitudes Surface Snowfall and Cloud Phase in ERA5 and CMIP6 using Satellite Observations" Hellmuth et al.**

**General Comments**

In the following response to the reviewer we keep the reviewers comments (black) and our answers from the previous review (dark blue). The responses to the current review phase are in light blue, additionally newly added text to the manuscript is in italic-bold.

The authors have addressed in detail all my comments and concerns from the initial submission. I commend their efforts and appreciate the additional clarity of the revised manuscript.

My main concern from the initial submission was in regard to the generalizations/conclusions presented. The authors have added more precise language to better limit the scope of their conclusions based on their results.

I am satisfied with the authors' responses to all my minor suggestions and most of my main suggestions. Below are follow-up comments on one of my main suggestions. I have left the original suggestion and authors' response for completeness, and indicated my follow-up comment with ***.

(5) Lines 451-453: "*In contrast, McIlhattan et al. (2017) showed that the CESM-LE underestimates the fLCC by ~ 17% and overestimates the fsnow by ~ 57% in the Arctic. However, since we utilize a different metric (sLCC instead of LCC), there is no reason to expect the model biases to be identical.*" From my understanding of the two metrics, fLCC and fsLCC wouldn't be likely to produce biases with the opposite sign in the high-latitudes (outside of perhaps summer) if the models contained similar cloud systems. It seems more likely that the difference between this study's biases and those found by and McIlhattan et al. (2017) arise from differences in the models' cloud systems. The same metrics from McIlhattan et al. (2017) were used to evaluate LCCs in CESM2 (a CMIP6 generation model; McIlhattan, E. A., et al. (2020). Arctic clouds and precipitation in the Community Earth System Model version 2. Journal of Geophysical Research: Atmospheres, 125, e2020JD032521. https://doi.org/10.1029/2020JD032521), and those findings appear to be more similar to those presented here. I strongly encourage the authors include the issue of model generation when comparing their results to earlier studies.

The following figures show the fLCC (a-d) as defined by McIllhattan et al. (2017) and the fsLCC (e-h) as defined in our study. The lower panel in the figures (i-l) show the difference between the two metrics. fLCC and fsLCC are especially different over the ocean.

[Figure]

We included the additional reference given by the reviewer and added some sentences to the discussion on model versions. Specifically, we added the following sentences in **Lines 492-509**:

"*However, since we utilize a different metric (sLCC instead of LCC), there is no reason to expect the model biases to be identical. Nevertheless, in a further study by McIlhattan et al. (2020) it was shown that the LCC frequency in the newer CESM version is more aligned with the satellite observations except for in the summer months, where it overestimates the LCC frequency.*

*Similarly, we generally observe that ERA5 and the CMIP6 mean overestimate the fsLCC to various extents during all seasons. These overestimations are likely linked to the microphysical parametrizations of cloud processes that govern cloud phase. This finding aligns with McIlhattan et al. (2020), indicating that while newer model versions have advanced in representing LCC frequencies more accurately, they can still overestimate cloud occurrences, particularly in specific seasons. Precipitation in the new CESM version is more frequent but lighter overall compared to the previous version, which is similar to our findings indicating that the models' sLCCs produce continuous snowfall, analogous to the "perpetual drizzle" problem (Mülmenstädt et al., 2020; Lavers et al., 2021).*

*Furthermore, while McIlhattan et al. (2017, 2020) considered a single ESM, our study considers an ensemble of CMIP6 models. Nevertheless, the insights from McIlhattan et al. (2017, 2020) provide relevant context for our finding that ERA5 and the CMIP6 model mean produce sLCCs more frequently than observed in the NH and SH mid-to-high latitudes, especially over the sea ice and land depending on the season (Figs. C1 and C2). In these regions, not only is the frequency of occurrence of sLCCs too high, but the sLCCs are too efficient at producing snowfall in ERA5 and CMIP6 models (Figs. 5 and 6). The latter finding is consistent with the findings of McIlhattan et al. (2017) that LCCs produce snow too frequently in the CESM-LE model.*"

*\*\*\*I appreciate these clarifications and additions. The two plots are very helpful in understanding the regional differences in the two metrics. I had not fully thought about the implications of the authors' requirement of FsLCCs having temperatures below freezing at the surface: it essentially removes from the analysis all clouds from most open ocean grid-boxes. This effect would be particularly strong in summer but important in other seasons as well.*

*Is it correct to reason that slightly colder model/reanalysis surface temperatures relative to CloudSat could result in FsLCC overestimations relative to observations – independent from differences in cloud representation? For example, if a given region had liquid cloud properties/frequency that were identical in ERA5 and CloudSat, but ERA5 had surface temperatures slightly below 0C and CloudSat slightly above, then would ERA5 have higher FsLCC for that region? If this is a correct interpretation, I would suggest modifying the language of the paper to avoid making strong conclusions about sLCCs over the open ocean. (e.g. the abstract notes "Specifically, we find that the ERA5 reanalysis and ten CMIP6 models consistently overestimate the frequency of sLCCs and snowfall frequencies from*

*sLCCs compared to CloudSat-CALIPSO satellite observations, especially over open ocean regions.")*

Thank you for raising this point, that differences in sLCC between ERA5 and CMIP6 models and CloudSat could also come from temperature differences. While our additional analyses (presented below, Figs. Rev. 1, Rev. 2, and Rev. 3) shows that this is generally not the case, a notable exception is the CMIP6 ensemble mean in the central Arctic in summer (Fig. 1), we thank the reviewer for requesting the additional analyses that made this evident. We have now removed "especially over open ocean regions" from the abstract, and have added a sentence in the results and a short discussion about this in the manuscript.

Lines 305 - 307: "*It is reasonable to assume that the temperature in the ECMWF-AUX product used in CloudSat-CALIPSO is quite similar to the ERA5 daily mean **as indicated by the seasonal mean 0°C isotherm in Figs. 1 and 2**. However, ERA5 shows a slight variation in the 0°C isotherm line over Central Europe during DJF compared to ECMWF-AUX (Fig. 1 e). Furthermore, a* comparison of the 2m temperature between ECMWF-AUX and ERA5 shows a latitudinal average difference of 0.24K ± 0.22K (Fig. D1)."*

Lines 540 - 548: "***For the most part, the 0°C isotherm shown in Figs. 1 and 2 support our argument that the primary issue with the ERA5 and CMIP6 datasets lies not with the simulated temperature itself, but with the representation of cloud properties and microphysics. This distinction highlights that the observed deviations in fsLCC and fsnow are driven more by inaccuracies in cloud simulation than by temperature discrepancies. Exceptions occur over Central Europe during DJF between ECMWF-AUX, ERA5 and the CMIP6 ensemble mean (Figs. 1 a, e, i). However, more notably is the difference in the CMIP6 ensemble mean over the central Arctic during summer (Fig. 1 k), where simulated temperatures appear to be too cold. In this specific case, the cloud bias could stem from a temperature bias, suggesting a potential link between temperature inaccuracies and cloud simulation in the CMIP6 ensemble mean for this region and season.***"*

As for the text highlighted by the reviewer (*"Specifically, we find that the ERA5 reanalysis and ten CMIP6 models consistently overestimate the frequency of sLCCs and snowfall frequencies from sLCCs compared to CloudSat-CALIPSO satellite observations, especially over open ocean regions."*), this simply states the facts, without attributing the bias to any particular cause such as cloud microphysics or temperature. In the previous review we introduced Figure D1, which displays the zonal mean 2m temperature of ECMWF-AUX (solid black line) and ERA5 (dashed orange line) for the mid-to-high latitudes. We noted that the temperature errors are less likely than those associated with microphysical variables. Further strengthening this argument, we observe minimal differences in grid cell 2m temperature correlation between ECMWF-AUX and ERA5 (shown in Figure Rev. 1), which both come from the same weather forecast centre. The correlation coefficients for these datasets are high ($R^2$=0.81°C for the Northern Hemisphere and 0.97°C for the Southern Hemisphere) when looking at the 0°C±2°C grid cell 2m temperature range. This suggests that discrepancies are likely due to cloud occurrence and not the applied temperature threshold. Moreover, the seasonal averages of fLCC for both hemispheres (Figs. Rev. 2 and Rev. 3) show that ERA5 and CMIP6 ensemble mean tend to overestimate the liquid containing cloud

occurrence, further supporting that the 2m temperature values are not the primary issue, as no temperature threshold was applied to obtain fLCC.

[Figure]

Figure Rev. 1: Correlation between grid cell 2m temperature from ECMWF-AUX and ERA5 for the Northern Hemisphere (NH, panels a and c) and Southern Hemisphere (SH, panels b and d), with colours indicating individual seasons. Panels (a) and (b) display correlation across the full temperature range for all grid cells, whereas panels (c) and (d) restrict the view to temperatures close to 0°C, excluding grid cells with temperatures above 2°C or below -2°C. The lower-right box in each plot shows the linear regression between ECMWF-AUX and ERA5 grid cell 2m temperatures and provides the correlation coefficient (R²) for 2007-2010, independent of the season, highlighting the strength of alignment between the two datasets.

[Figure]

Figure Rev. 2: Seasonal averages of fLCC in the NH mid-to-high latitudes between 2007 and 2010. Combined CloudSat and CALIPSO observations are shown in the first row (a-d). The last two rows are the difference plot. They are CloudSat-CALIPSO (CC) observations minus ERA5 (e-h) or CMIP6 model mean (i-l) where valid data occurs, with green (pink) values showing underestimation (overestimation) in ERA5 and the CMIP6 model mean concerning the satellite observations. Areas where the difference between CloudSat-CALIPSO and CMIP6 model mean is not significant (< 95%) are marked with hatches. The area-weighted averages for the study area where CloudSat-CALIPSO has observations are displayed in the lower-left corner of each map. The black dashed line represents the seasonal mean 2m temperature 0°C isotherm for each individual product. The red line (in a-d) shows the average sea ice edge of 20% sea ice concentration (SIC) between 2007 and 2010, for the given season.

[Figure]

Figure Rev. 3: Seasonal averages of fLCC in the SH mid-to-high latitudes. Layout and differences are identical to Figure Rev. 2.

*It would also be very helpful to include mean spatial areas where T2m < 0C for the subplots in Figures 1&2 and/or the charts in Figure 3. Ideally these areas would be similar across the obs/reanalysis/model datasets. If the areas are much larger in the reanalysis and models, that would be an important contributing factor to include in the discussion of the area weighted mean FsLCC values in Figures 1&2&3.*

In light of this and the previous comment, we have now revised Figures 1, 2, 5, and 6 in the manuscript, and added the seasonal mean 2m temperature 0°C isotherm on these Figures to show the areas with 2m temperatures below 0°C.

[Figure]

**Figure 1.** The NH mid-to-high latitude seasonal averages of fsLCC. The first row (a-d) for CloudSat-CALIPSO, the second row (e-h) displays ERA5 data, and the third row (i-l) shows the CMIP6 model mean. Each map includes an area-weighted average for the study area (lower left corner). These averages are calculated for areas where CloudSat-CALIPSO have valid observations (between 45°N − 82°N) and exclude the dotted area (in e-l). **The black dashed line represents the seasonal mean 2m temperature 0°C isotherm for each individual product.** The red line (in a-d) shows the average sea ice edge of 20% sea ice concentration (SIC) between 2007 and 2010, for the given season.

[Figure]

**Figure 2.** The SH mid-to-high latitude seasonal averages of fsLCC. The first row (a-d) for CloudSat-CALIPSO, the second row (e-h) displays ERA5 data, and the third row (i-l) shows the CMIP6 model mean. Each map includes an area-weighted average for the study area (lower left corner). These averages are calculated for areas where CloudSat-CALIPSO have valid observations (between 45°S − 82°S) and exclude the dotted area (in e-l). **_The black dashed line represents the seasonal mean 2m temperature 0°C isotherm for each individual product._** The red line (in a-d) shows the average sea ice edge of 20% sea ice concentration (SIC) between 2007 and 2010, for the given season.

[Figure]

**Figure 5.** The figure presents the seasonal averages of fsnow in the NH mid-to-high latitudes. The layout and area-weighted averages are calculated the same as those shown in Figure 1.

[Figure]

**Figure 6.** The figure presents the seasonal averages of the fsnow in the SH study region. The layout and area-weighted averages are calculated the same as those shown in Figure 1.